# Examining ENSO related variability in tropical tropospheric ozone in the RAQMS-Aura chemical reanalysis

Maggie Bruckner[1], R. Bradley Pierce[1,2], Allen Lenzen[2]

[1]Department of Atmospheric and Oceanic Sciences, University of Wisconsin-Madison, Madison, WI, 53706, USA
[2]Space Science and Engineering Center, University of Wisconsin–Madison, Madison, WI, 53706, USA

*Correspondence to*: Maggie Bruckner (mebruckner@wisc.edu)

**Abstract**

The El Niño-Southern Oscillation (ENSO) is a major driver of interannual variability in both tropical and mid-latitudes and has been found to have a strong impact on the distribution of tropospheric ozone in the tropical Pacific in satellite observational
datasets, chemical transport models, and chemistry-climate simulations. Here we analyze inter-annual variability in tropical tropospheric ozone by applying composite analysis, empirical orthogonal function (EOF) analysis and multiple linear regression to the Real-time Air Quality Modeling System (RAQMS) Aura chemical reanalysis. As shown in similar studies, the dominant mode of inter-annual variability in tropical tropospheric ozone is driven by ENSO. ENSO composites show that the ENSO signature in tropospheric ozone is strongest near the tropopause. We also show an enhancement in tropical ozone
over the maritime continent below 700 hPa during El Niño that is dependent on the magnitude of the biomass burning emissions in the region. We reconstruct the ENSO variability in tropical tropospheric ozone through a multiple linear regression of principal components for precipitation and CO. The multiple linear regression quantifies that variability in biomass burning contributes to ENSO variability in tropical tropospheric ozone though the dominant driver is convective precipitation.

## 1 Introduction

The development of methods to calculate tropospheric ozone residuals (TOR) from satellite total column observations (eg. Fishman and Balok, 1999; Fishman et al., 1990; Fishman and Larsen, 1987) provided the first global view of tropospheric ozone and showed a systematic zonal wave one structure in the tropics. This zonal wave one structure is consistent with the climatological average state of tropical atmosphere, which is dominated by the Pacific Walker circulation, defined by ascending motion over warm SSTs near the maritime continent and descending over cooler SSTs in the eastern Pacific, with
easterlies at surface and westerlies aloft. Climatologically, tropospheric ozone columns are lowest over the Pacific and highest downwind of western Africa (Fishman et al., 1990, 1996, 2003). The enhancement downwind of western Africa is strongest during September-October-November (SON) and is associated with photochemical production of ozone from biomass burning emissions (Fishman et al., 1996, 2003, 2005). Tropospheric ozone concentrations over Africa and South America are lowest in March-April-May (MAM) (Fishman et al., 1990, 2003). The Fishman, Wozniak, and Creilson 2003 TOR seasonal

climatology also shows a variance of 5-10 DU over the maritime continent from December-January-February (DJF) to June-July-August (JJA). The El Niño-Southern Oscillation (ENSO) is a major driver of inter-annual variability in both tropical and mid-latitudes (eg. McPhaden et al., 2006; Trenberth, 1997), and has been found to have a strong impact on the distribution of tropospheric ozone in the tropical Pacific (Doherty et al., 2006; Peters et al., 2001; Sekiya and Sudo, 2012; Ziemke et al., 2010).

ENSO phases of El Niño and La Niña are tracked using a variety of indexes including the Niño 3.4 index (Bamston et al., 1997; Trenberth, 1997) and the Ozone ENSO Index (Ziemke et al., 2010). El Niño events occur when a warm SST anomaly develops in the eastern Pacific and reduces the east-west temperature gradient across the equatorial Pacific. In response to the SST anomaly, the trade winds weaken. Convection is enhanced over the eastern Pacific, leading to increased precipitation in the region and an eastward shift of the Walker Circulation. Correspondingly convection is suppressed over the maritime

continent and leads to drier than usual conditions. During El Niño events, tropospheric ozone is lower over the Pacific as the enhanced convection lofts low ozone air masses from near the ocean surface higher into the column, and higher over the maritime continent as higher upper tropospheric ozone concentrations descend (eg. Doherty et al., 2006; Hou et al., 2016; Sudo and Takahashi, 2001). Variability in the location of the maximum SST anomaly during the El Niño phase has led to a distinction between canonical (eastern Pacific) El Niño events and El Niño Modoki (central Pacific) events (eg. Larkin and Harrison,

2005; Kim and Yu, 2012; Santoso et al., 2017). In the canonical El Niño, the maximum SST anomaly extends into the eastern tropical Pacific cold pool while during El Niño Modoki the maximum SST anomaly is in the central Pacific. The ascending branches of the Walker circulation are over the central Pacific during El Niño Modoki (Ashok et al., 2007). Following from the differences in the Walker circulation, the pattern of the ENSO response in tropical tropospheric ozone depends on the type of El Niño (Hou et al., 2016).

La Niña events occur when the eastern Pacific is cooler than average, and the atmosphere responds in a generally opposite, though not symmetric, manner to El Niño as enhanced vertical motion and convection occurs over the maritime continent, suppression of convection occurs over the east Pacific, and enhanced downwelling over the east Pacific. Tropical tropospheric ozone columns reflect the impacts of  higher upper tropospheric ozone concentrations descending over the Pacific and comparatively lower concentration lower tropospheric ozone  ascending near the maritime continent during La Niña (eg.

Ziemke and Chandra, 2003; Doherty et al., 2006).

     The influence of ENSO on tropospheric ozone has previously been investigated in observational datasets, chemical transport models, and chemistry-climate models. Application of statistical techniques (regression, correlation, and empirical orthogonal functions) to TOR data revealed that interannual variability in measurements over the tropical Pacific is dominated by ENSO (eg. Doherty et al., 2006; Oman et al., 2013; Ziemke et al., 1998, 2010). ENSO variability in tropical tropospheric ozone

columns has been reproduced in chemical transport models and climate models (eg. Sudo and Takahashi, 2001; Chandra et al., 2002; Peters et al., 2001; Doherty et al., 2006; Sekiya and Sudo, 2014). ENSO variability in equatorial Pacific tropospheric ozone was initially thought to be equally due to shifts in biomass burning emissions and meteorological conditions (Chandra et al., 2002; Sudo and Takahashi, 2001). More contemporary studies indicate enhancement in biomass burning during El Niño

results in regional enhancement of ozone with little contribution to global tropospheric ozone variability and that the response

of tropospheric ozone to ENSO is primarily due to dynamical processes (Doherty et al., 2006; Inness et al., 2015).

In this study, we will investigate the inter-annual variability of tropical tropospheric ozone in a chemical re-analysis extending from 2006 through 2016. A chemical re-analysis produces a long-term data record by cycling  a model forecast and data assimilation system to combine forecasts and observations in a statistically consistent manner that accounts for forecast and observation error (Miyazaki et al., 2020; Yumimoto et al., 2017). The data record obtained is a best-estimate of the real

composition of the atmosphere, as analyses produced are constrained by observations of a limited number of species and the evolution of those species by model physics (Miyazaki et al., 2020). A comparison of several recent chemical re-analyses including the Copernicus Atmospheric Monitoring Service (CAMS) reanalysis (Inness et al., 2019), and the Tropospheric Chemistry Reanalysis version 2 (TCR-2) (Miyazaki et al., 2020) found that these analyses are suitable for generating ozone climatologies and looking at trends, though individual re-analyses will differ due to model configuration (Huijnen et al., 2020).

While chemical re-analysis has been used to look at the ENSO signal in CO, $O_3$, $NO_x$, and smoke aerosols (Inness et al., 2015), our analysis will make use of the chemical production and loss terms, convective mass flux, and diabatic heating from a chemical re-analysis to examine variability in tropospheric ozone. We also focus on the 2006-2016 period, which includes significant biomass burning events during the 2015/2016 El Niño event.

This study seeks to: 1) evaluate the tropical tropospheric ozone column variability associated with ENSO in a 1x1 degree

chemical re-analysis using the Real-time Air Quality Modeling System (RAQMS, Pierce et al., 2007) and satellite measurements from the NASA Aura satellite (Pierce et al., 2016) and 2) investigate how the 2015/2016 extreme El Niño event impacts the ENSO response.

## 2 Methods

### 2.1 RAQMS-Aura

The Real-time Air Quality Modeling System (RAQMS) Aura Reanalysis, hereafter RAQMS-Aura, is a chemical re-analysis using RAQMS (Pierce et al, 2007), a global chemical transport model with full stratospheric and tropospheric chemistry, and satellite trace gas and aerosol retrievals from the NASA satellites (Terra, Aqua, and Aura) covering 2006 through 2016. RAQMS-Aura provides 1°x1° global chemical analyses, on 35 hybrid model levels from the surface to approximately 60 km above ground level, at 3-hour time steps. The operational grid point statistical interpolation (GSI) 3-dimensional variational

analysis system (Wu et al., 2002) is used to assimilate retrievals from the following Aura instruments: Aura Ozone Monitoring Instrument (OMI) cloud cleared total column ozone (McPeters et al., 2008), Microwave Limb Sounder (MLS) (Froidevaux et al., 2008) stratospheric ozone profiles, and OMI tropospheric column $NO_2$ (Boersma et al., 2007; Bucsela et al., 2013). NASA Terra and Aqua Moderate Resolution Imaging Spectrometer (MODIS) aerosol optical depth (AOD) (Remer et al., 2005) and Atmospheric Infrared Sounder (AIRS) carbon monoxide profile (Maddy and Barnet, 2008; McMillan et al., 2005; Yurganov

et al., 2008) are also assimilated at three-hour intervals. Analysis increments from the OMI tropospheric column $NO_2$ retrievals

are used for off-line adjustment of apriori 2010 Hemispheric Transport of Air Pollution (HTAP, 2010) anthropogenic emission inventories following an offline mass balance approach similar to East et al. 2022. Biomass burning emissions in RAQMS-Aura use Terra and Aqua MODIS fire detections and are calculated using a bottom-up approach developed by Soja et al. 2004 and compared to other approaches in Al-Saadi et al. 2008. This approach estimates total carbon emissions at MODIS fire detections with the US Forest Service Haines Index (Haines, 1989) to determine fire weather severity and gridded, ecosystem-dependent estimates of carbon consumption for low, medium, and high fire severity fires. Emission ratios are then used to estimate emissions of CO, $NO_x$, and hydrocarbons from the calculated total carbon emissions.

The dynamical core of RAQMS is the UW hybrid model (Schaack et al., 2004). The UW hybrid model utilizes physical parameterizations from the NCAR Community Climate Model (CCM3) (Kiehl et al., 1998), including the moist convection scheme. The CCM3 moist convection scheme combines the Zhang and McFarlane (1995) deep convection scheme with shallow and midlevel convection following Hack (1994). The deep convection scheme treats convection as an ensemble of updrafts and downdrafts, and the shallow convection scheme treats convection as separate plumes within 3 successive layers whereby mass is detrained from one layer into the next (Kiehl et al., 1998; Zhang et al., 1998). RAQMS-Aura initializes its meteorological fields with archived analyses from the National Center for Environmental Prediction (NCEP) Global Data Assimilation System (GDAS) (Kleist et al., 2009; Wang et al., 2013). These fields are impacted by updates to physics, resolution, and data assimilation used in the GDAS system (MODEL CHANGES SINCE 1991, 2023).

**2.2 ENSO Composites**

Anomaly composites are used to evaluate how well RAQMS-Aura reproduces observed ENSO variability. El Niño and La Niña periods are determined by use of the Niño 3.4 index. ENSO events are defined as occurring when the index is at least 0.4°C greater (El Niño) or less (La Niña) than average for 5 consecutive months (eg. Trenberth, 1997; Ziemke et al., 2015). Anomalies are defined as the deviation from the average annual cycle during the RAQMS-Aura analysis period (2006-2016). Anomaly composites for El Niño and La Niña periods are generated for precipitation, convective mass flux, diabatic heating, ozone concentration, carbon monoxide, and net ozone production from monthly mean RAQMS-Aura analyses. Anomaly composites are also generated for satellite observations of tropospheric ozone column, total column carbon monoxide, and total precipitation. To investigate the vertical structure of ENSO variability in RAQMS-Aura, anomaly cross section composites are calculated between 7.5°S to 2.5°N for convective mass flux, diabatic heating, ozone, carbon monoxide, and net ozone production.

**2.3 Empirical Orthogonal Function (EOF) Analysis**

EOF analysis has been used previously by Peters et al. 2001 and Doherty et al. 2006 to identify ENSO variability in modeled tropospheric ozone concentrations. EOF analysis is performed on de-seasonalized and de-trended precipitation, CO column, and tropical tropospheric ozone column (TTOC) monthly mean anomalies to determine the dominant modes of tropical variability in RAQMS-Aura analyses.

Following Doherty et al. 2006 the resulting EOF patterns for each RAQMS-Aura variable are multiplied by the standard deviation of the associated principal component (PC) to produce the physical magnitude of change associated with the mode. The PCs are correlated against the Niño 3.4 index to assess whether the mode captured by the EOF accounts for ENSO variability. A multiple linear regression is constructed using the precipitation and CO PCs to investigate how variability in convection and biomass burning emissions drive the ozone ENSO signal.

## 3 Results

### 3.1 Validation of RAQMS-Aura Precipitation

Prior to investigating variability of the RAQMS-Aura chemical fields, we evaluate RAQMS-Aura convection and precipitation processes through comparisons with observations. In RAQMS-Aura, sub-grid-scale mass flux between model layers occurs through shallow and deep convective schemes. Diabatic heating is generated by the sub-grid-scale convective parameterizations and influences the grid-scale thermodynamics. Convective mass flux and diabatic heating will be used in the composite analysis to look at the impact of ENSO on vertical transport and tropical tropospheric ozone concentrations.

Monthly mean total and convective precipitation from RAQMS-Aura is compared to estimates of precipitation from the Tropical Rainfall Measuring Mission (TRMM) Multi-satellite precipitation Analysis (TMPA) 3B43 product (Huffman et al., 2007). TRMM 3B43 merges satellite IR and microwave precipitation estimates with rain gauge data to produce a best estimate of monthly mean precipitation rate from 50°S to 50°N at 0.25x0.25 degree resolution, which in this study is averaged onto the RAQMS 1x1 degree grid. Our analysis is focused on meridional structure and seasonal maps to look at average regional biases, and time-series of the maritime continent and Pacific Intertropical convergence zone (ITCZ) regions to look at longer-term trends.

### 3.1.1 Meridional Structure

Figure 1 displays the meridional averaged convective, large-scale, and total precipitation for RAQMS-Aura and total precipitation from TRMM 3B43 for each season. The seasonal average meridional precipitation maxima in RAQMS-Aura are broader than observed in TRMM 3B43. During DJF and MAM, observed tropical precipitation peaks in both the northern hemisphere (NH) and southern hemisphere (SH). During JJA and SON, observed tropical precipitation peaks only in the NH. In DJF the observed hemispheric peaks are of similar magnitude with the NH peaking at 0.247 mm/hour and the SH peaking at 0.233 mm/hour. TRMM 3B43 MAM indicates that the NH branch is more active during this season than the SH branch, as the NH peak is 0.293 mm/hour, and the SH peak is 0.229 mm/hour. RAQMS-Aura reproduces the observed double peaks for DJF and MAM, though the magnitude is overestimated in RAQMS-Aura by 0.08-0.12 mm/hour, and the DJF SH peak is larger than the NH peak and 5 degrees to the south of the observed peak. In JJA and SON, the reanalysis reproduces the observed single maxima, though it is broader by more than 15 degrees latitude, and the absolute maximum is displaced approximately 2.5 degrees to the north.

Between 40°N and 40°S the total precipitation in RAQMS-Aura is predominately convective precipitation, with ratios of convective precipitation to total precipitation exceeding 0.6 on average. It is common for tropical precipitation to be predominately convective precipitation in global models, leading to a "drizzling bias". This "drizzling bias" is the result of convective parameterizations producing convective precipitation that is too frequent and long-lasting but not as intense as observed while the total precipitation amount is realistic (Chen et al., 2021; Chen and Dai, 2019).

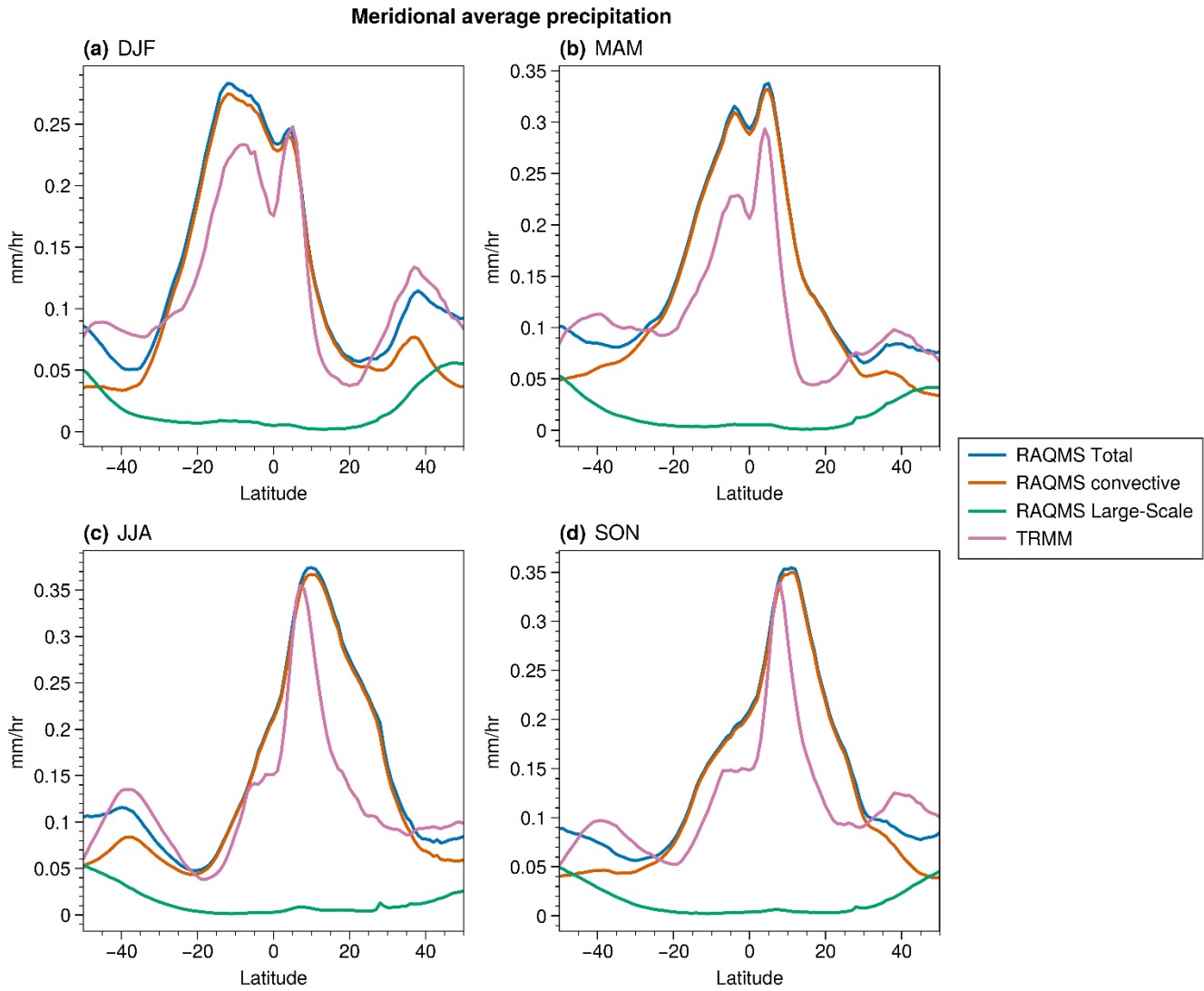

**Figure 1. Zonally and seasonally averaged precipitation from RAQMS-Aura and TRMM 3B43 for a) DJF, b) MAM, c) JJA, and d) SON.**

### 3.1.2 Horizontal Structure

While RAQMS-Aura reasonably reproduces the seasonality of the observed meridional structure, the distributions are broader than in observations. Seasonal maps of precipitation allow us to examine the reasons for this in more detail. Figure 2 shows seasonal maps of precipitation from the TRMM 3B43 observations and RAQMS-Aura. TRMM 3B43 and RAQMS-Aura are well correlated for all seasons, with DJF displaying a spatial correlation of 0.86, MAM a spatial correlation of 0.75, JJA a spatial correlation of 0.71, and SON a spatial correlation of 0.77. These correlations show that the RAQMS-Aura reanalysis broadly captures the seasonal changes in the spatial pattern of tropical precipitation.

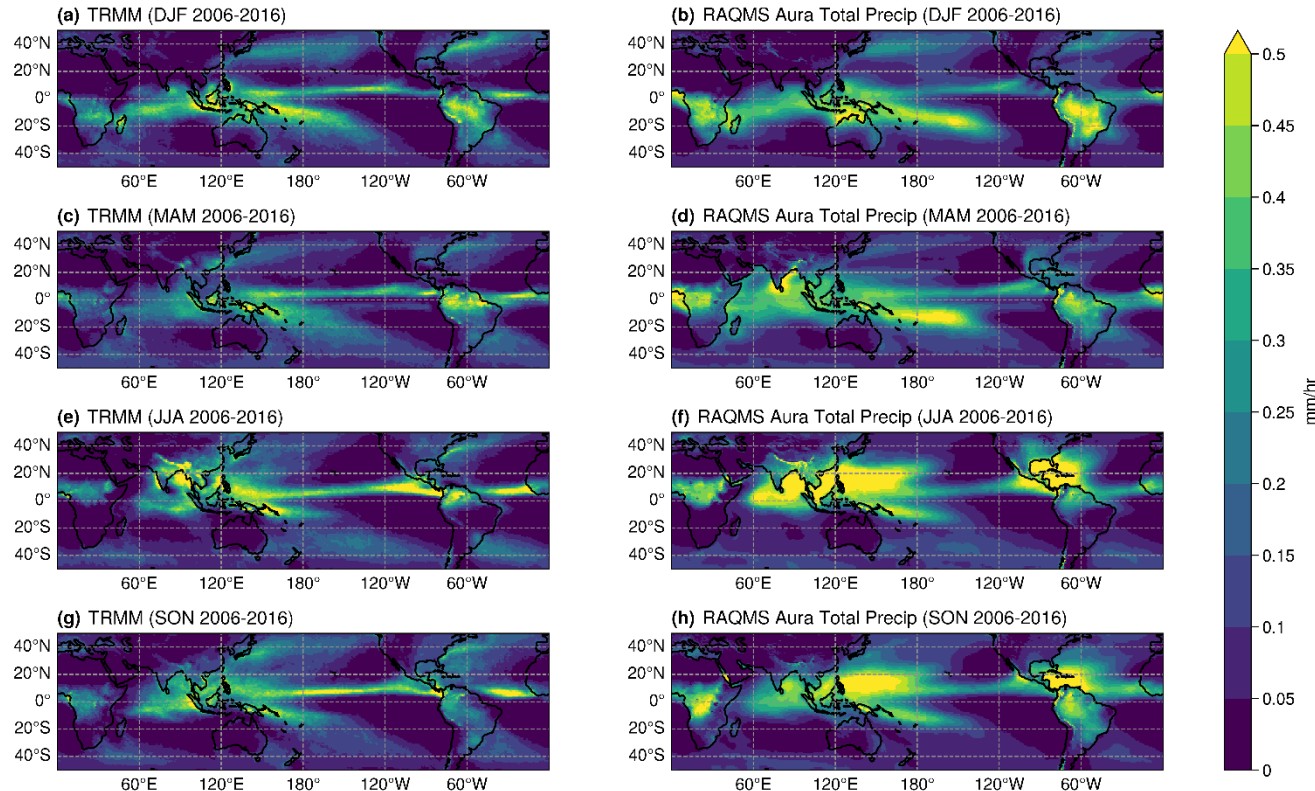

**Figure 2. Seasonal mean precipitation for TRMM 3B43 (a, c, e, g) and RAQMS-Aura (b, d, f, h).**

Precipitation over land in South America and Africa is consistently overestimated relative to TRMM 3B43 by 0.2-0.3 mm/hour. This overestimation over land is a long-standing bias of the dynamical component of RAQMS (Schaack et al, 2006). RAQMS-Aura overestimates precipitation in the Gulf of Mexico and Caribbean by >0.3 mm/hour during JJA and SON. During DJF and MAM, the average bias over the Gulf of Mexico is less than +/- 0.1 mm/hour. RAQMS-Aura overestimates precipitation over the Caribbean by ~0.14 mm/hour during DJF and by ~0.16 during MAM. RAQMS-Aura overestimates precipitation near India by >0.3 mm/hour during MAM and JJA. In the northwest Pacific, RAQMS-Aura shows larger overestimates of

precipitation in JJA and SON relative to DJF and JJA, with overestimates relative to TRMM of 0.05 mm/hour in DJF, >0.3 mm/hour in JJA, 0.15 mm/hour in MAM, and >0.3 mm/hour in SON.

RAQMS-Aura does capture precipitation features like the ITCZ and western North Atlantic storm track well, though there is bias in the precipitation amount. RAQMS-Aura underestimates precipitation in the western North Atlantic off the east coast of the US along a storm track region by 0.17 mm/hour in DJF, ~0.15 mm/hour in JJA, ~0.15 mm/hour in MAM, and ~0.17 mm/hour in SON. During DJF, precipitation is overestimated by 0.2-0.3 mm/hour in RAQMS-Aura in the Southern Hemisphere maximum over the Pacific and off the northern coast of Australia. The strength of the SH maximum is consistently overestimated by RAQMS-Aura, as it is higher than TRMM 3B43 by ~0.1 mm/hour in JJA, 0.25-0.3 mm/hour in MAM, and ~0.1 mm/hour in SON. RAQMS-Aura tends to underestimate the strength of the ITCZ in all seasons, with a small underestimate of ~0.05mm/hour in MAM and ~0.15 mm/hour in DJF. RAQMS-Aura underestimates the ITCZ over the east and central Pacific by a max of ~0.25 mm/hour in SON and JJA.

### 3.1.3 Time Series

The comparison of TRMM 3B43 precipitation and RAQMS-Aura indicates that RAQMS-Aura captures the expected seasonality in the ITCZ and over landmasses though tends to overestimate convective precipitation. Following this characterization of regional biases in RAQMS-Aura, we look closer at how the RAQMS-Aura represents precipitation within the tropics by evaluating the time series for 3 key regions, which are defined in figure 3. The region over the maritime continent is defined by broadscale ascent in the average Walker Circulation. Time series for the maritime continent, NH ITCZ, and SH maximum regions are displayed in figure 4.

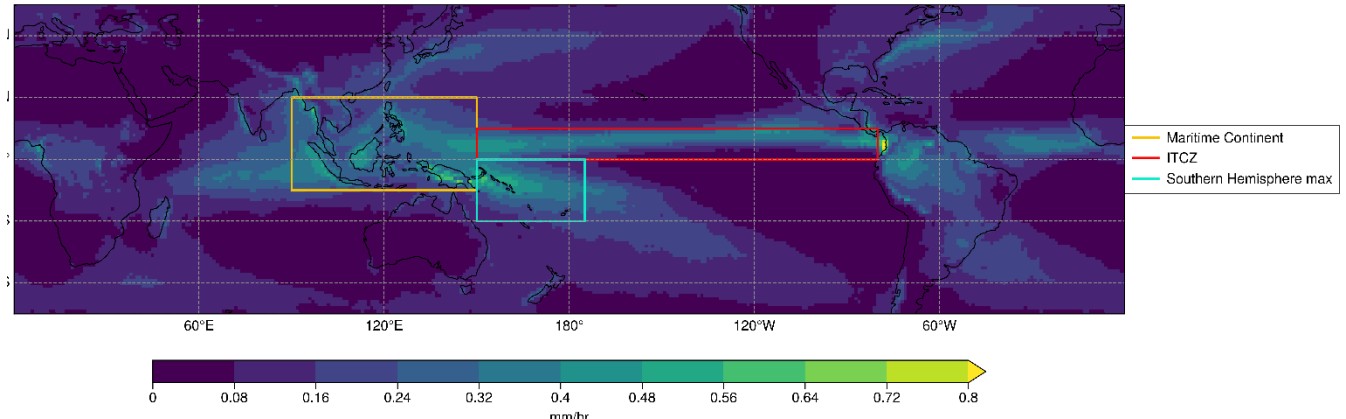

**Figure 3. Regions for timeseries overlaid on mean 2006-2016 TRMM precipitation.**

Over the maritime continent, RAQMS-Aura has a temporal correlation of 0.619 with TRMM and a mean bias of 0.064 mm/hour (22.27%). The bias between TRMM and RAQMS-Aura is initially higher, ~0.2 mm/hour at a max, then decreases after 2010 within this region. There is also an increased bias in 2015 and late 2016 over the maritime continent. Across the

ITCZ in the northern hemisphere RAQMS-Aura has a temporal correlation of 0.715 and bias of -0.0115 mm/hour (-4.90%) with TRMM. Prior to 2010 RAQMS-Aura displays a small bias relative to TRMM 3B43. Post 2010 RAQMS-Aura underestimates peak precipitation, though the temporal correlation of the measurements with TRMM 3B43 slightly increases to 0.774 within this region. Within a section of the SH precipitation maximum, RAQMS-Aura has a temporal correlation of 0.599 and bias of 0.038 mm/hour (13.53%) with TRMM. The good correlation and bias of less than 25% for each region

indicate that RAQMS-Aura has skill in reproducing the observed precipitation in the regions of interest for this study. Shifts in bias observed between 2009 and 2011 appear to be associated with upgrades to the GDAS system. Changes to GDAS implemented in 2009 included use of variational quality control in the assimilation system and flow dependent reweighting of background error variance (MODEL CHANGES SINCE 1991, 2023).

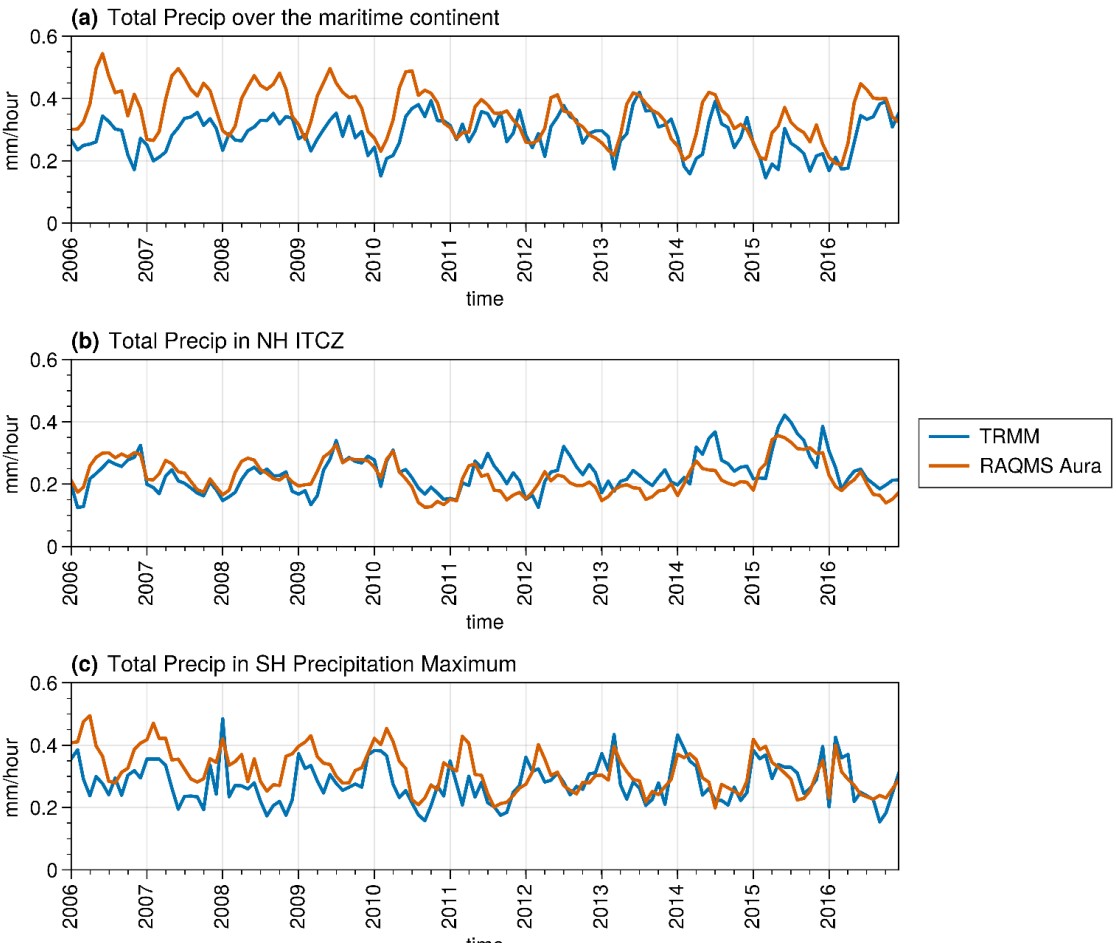

**Figure 4. Mean precipitation for TRMM 3B43 and RAQMS-Aura Precipitation over the maritime continent (a), in the NH ITCZ region (b), and in the SH maximum precipitation region (c). Over the maritime continent, RAQMS-Aura precipitation is on average biased 0.064 mm/hour (22.27%) higher than TRMM 3B43. In the NH ITCZ region RAQMS-Aura precipitation is on average biased 0.012 mm/hour (4.90%) lower than TRMM 3B43. In the SH maximum precipitation region RAQMS-Aura precipitation is on average biased 0.038 mm/hour (13.53%) higher than TRMM 3B43.**

## 3.2 Validations of RAQMS-Aura O$_3$ and CO

To establish fidelity of the RAQMS-Aura chemical fields, we evaluate ozone profiles, tropospheric ozone column, and CO column. The RAQMS-Aura monthly mean tropospheric ozone column is compared to the OMI-MLS TOR (Ziemke et al., 2006). The OMI-MLS TOR is a satellite residual product where total ozone columns from the OMI instrument and stratospheric columns from MLS instrument (both on-board the Aura satellite) are combined to infer the tropospheric ozone column. Monthly mean CO column from RAQMS-Aura is compared to CO column retrievals from Measurements of Pollution in the Troposphere (MOPITT) (Emmons et al., 2004). Both the OMI-MLS TOR and the MOPITT CO data used are monthly mean Level 3 products. We evaluate the RAQMS-Aura tropical O$_3$ vertical profiles with observations from 12 sites in the Southern Hemisphere Additional Ozonesondes (SHADOZ) network (Sterling et al., 2018; Thompson et al., 2017; Witte et al., 2017, 2018).

### 3.2.1 Horizontal Structure in CO and tropospheric O$_3$ columns

Seasonal maps of CO column and tropospheric ozone column are evaluated for RAQMS-Aura and satellite datasets. Figure 5 shows seasonal maps of CO columns from MOPITT and RAQMS-Aura. MOPITT and RAQMS-Aura are well correlated for all seasons, as DJF has a spatial correlation of 0.945, MAM a spatial correlation of 0.955, JJA a spatial correlation of 0.911, and SON a spatial correlation of 0.919. South American CO columns are overestimated in RAQMS-Aura by 0.4-0.8 x 10$^{18}$ mol/cm$^2$ in SON and 0.4-0.5 x 10$^{18}$ mol/cm$^2$ in JJA, and < 0.3 x 10$^{18}$ mol/cm$^2$ during DJF and MAM. Over the maritime continent, bias is < $\pm$ 0.2 x 10$^{18}$ mol/cm$^2$ during DJF, MAM, and JJA and biased low during SON by ~0.3 x 10$^{18}$ mol/cm$^2$. Over the Pacific, RAQMS-Aura has a high bias of 0.15-0.3 x 10$^{18}$ mol/cm$^2$ (< 25% difference).

Figure 6 shows seasonal maps of Tropospheric O$_3$ columns from OMI-MLS and RAQMS-Aura. OMI-MLS and RAQMS-Aura are well correlated for all seasons, as DJF has a spatial correlation of 0.822, MAM a spatial correlation of 0.995, JJA a spatial correlation of 0.934, and SON a spatial correlation of 0.941. While the correlations are strong, RAQMS-Aura tropospheric O$_3$ is consistently biased high by >2DU in the tropics relative to OMI-MLS.

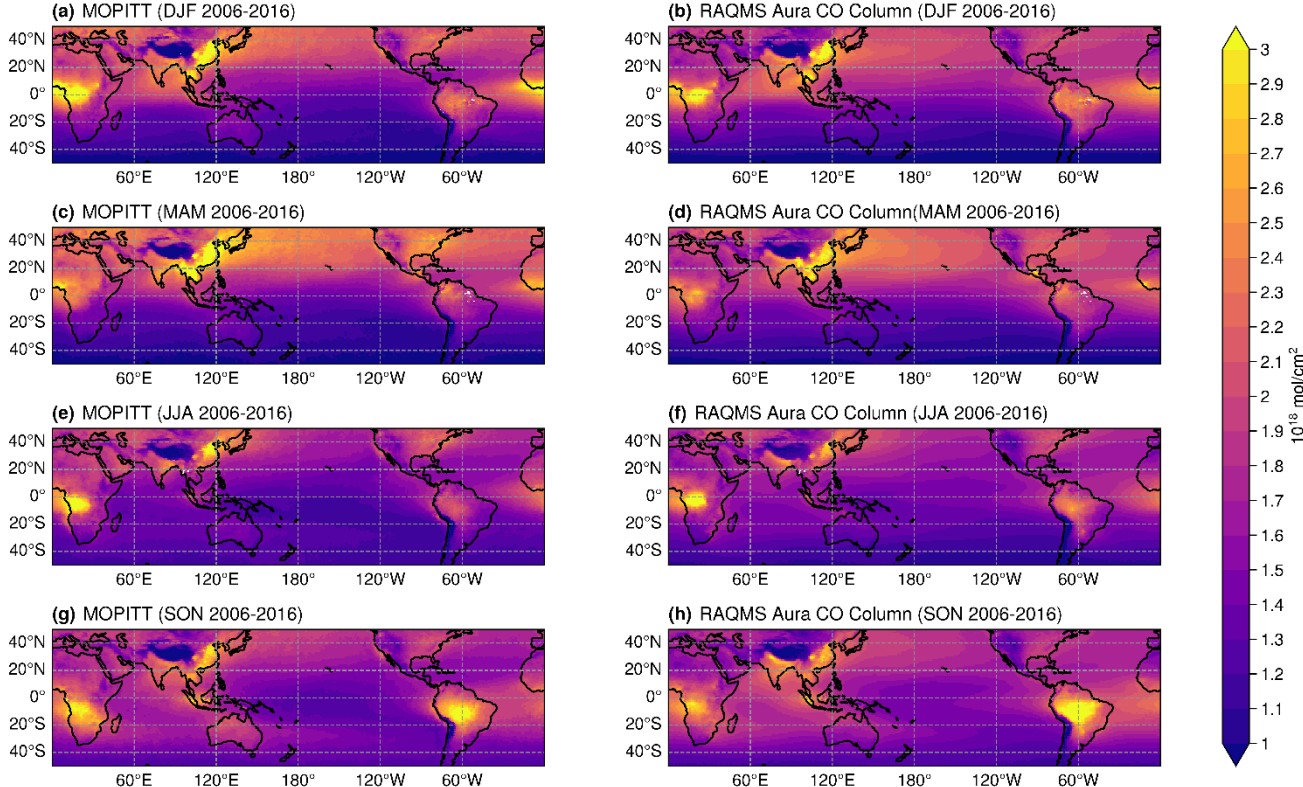

**Figure 5. Seasonal mean CO column for MOPITT (a, c, e, g) and RAQMS-Aura (b, d, f, h).**

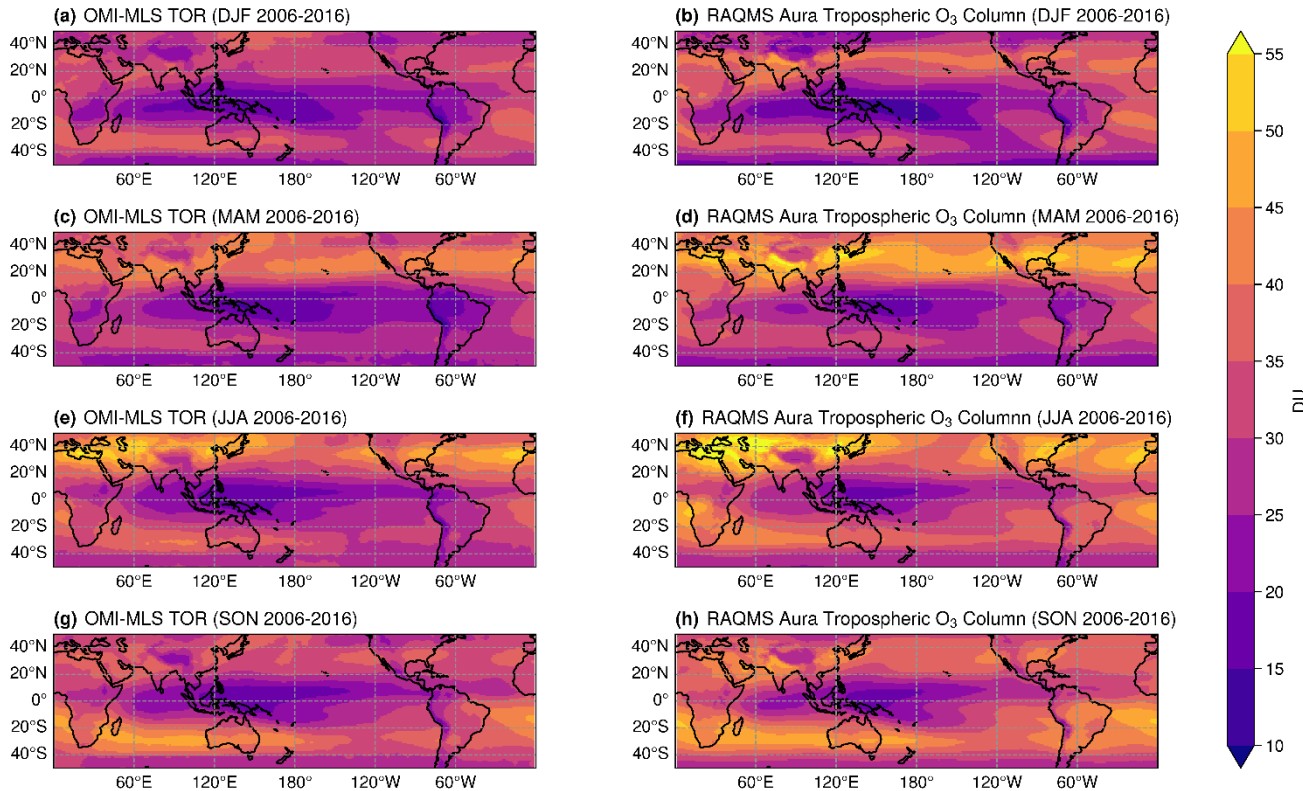

**Figure 6. Seasonal mean tropospheric O₃ column for OMI-MLS (a, c, e, g) and RAQMS-Aura (b, d, f, h).**

### 3.2.2 Time series of CO column and tropospheric O₃ column over the Maritime Continent

Following the characterization of seasonal mean regional biases in RAQMS-Aura CO column and tropospheric O₃ column, we look at how well RAQMS-Aura represents variability over the maritime continent (as defined in fig. 3). Timeseries of CO column and tropospheric O₃ over the maritime continent are displayed in figure 7. Unlike in the precipitation fields, the RAQMS-Aura CO columns and tropospheric O₃ columns do not exhibit a large shift in the bias over time.

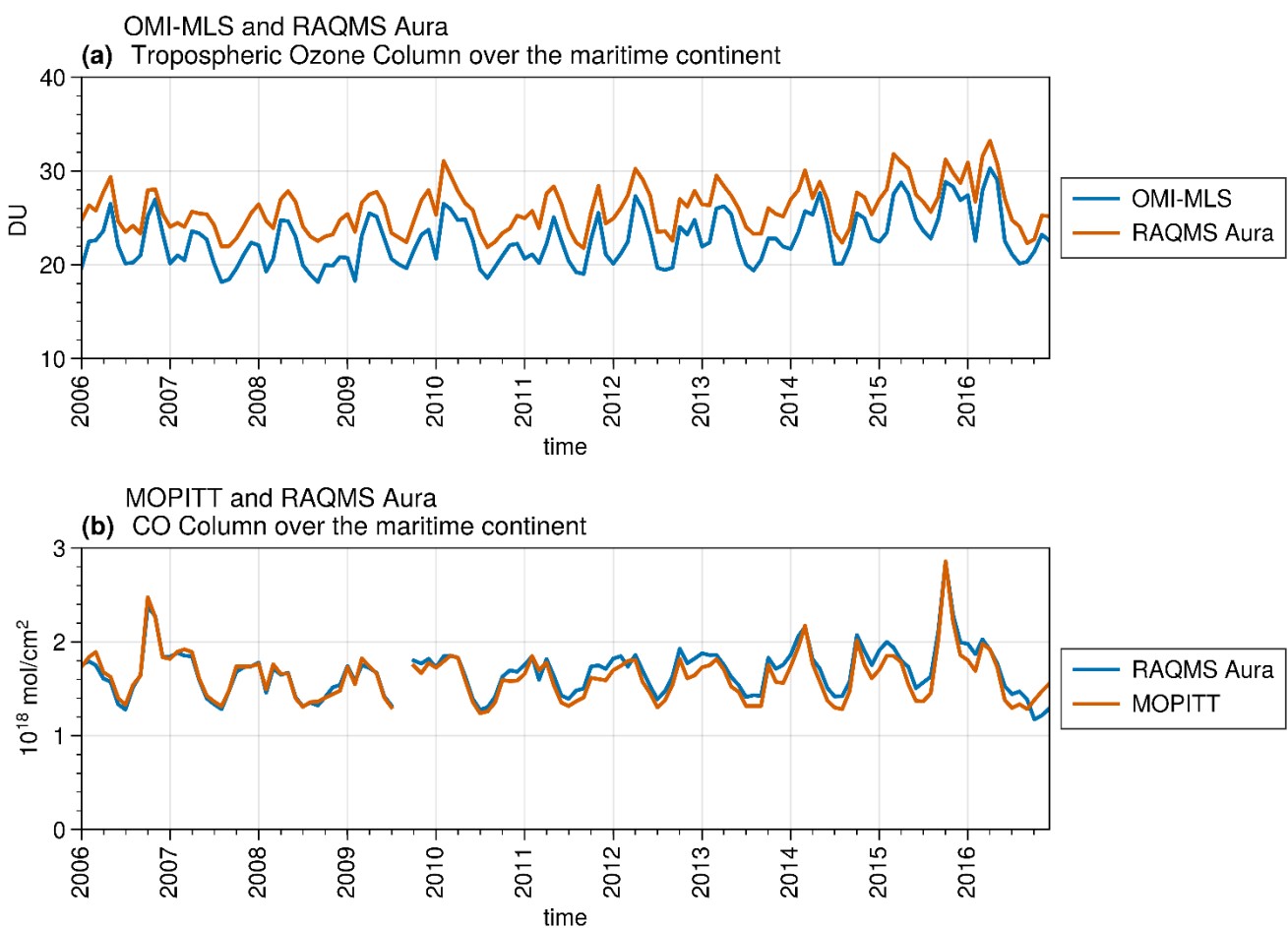

**Figure 7. Time series of mean tropospheric O₃ column (a) and CO column (b) over the maritime continent for RAQMS-Aura, MOPITT CO, and OMI-MLS TOR.**

RAQMS-Aura mean maritime continent tropospheric $O_3$ column has a temporal correlation of 0.937 with the OMI-MLS TOR and a mean high bias of 3.273 DU (14.435%). RAQMS-Aura mean maritime continent CO column has a temporal correlation of 0.943 with MOPITT and a mean high bias of 0.0477 x $10^{18}$ mol/cm² (2.93%). The very good temporal correlation and bias of less than 25% for both CO column and tropospheric $O_3$ column indicates that RAQMS-Aura has skill in reproducing the observed CO column and tropospheric $O_3$ column in a key region of interest for this study.

### 3.2.3 Vertical Structure of O₃

RAQMS-Aura ozone profiles are compared to the reprocessed v06 Southern Hemisphere ADditional OZonesondes (SHADOZ) ozone profiles (Thompson et al., 2021) at the SHADOZ sites in 100m altitude bins from 0km to 30km. The SHADOZ sites used in this study are shown in Figure 8 along with the 2006-2016 mean tropospheric ozone column from

RAQMS-Aura. The vertical distribution of mean bias in RAQMS-Aura $O_3$ profiles for all SHADOZ sites is presented in figure 9. RAQMS-Aura $O_3$ exhibits a high bias of >20% near the surface. Above 3km, the average bias in RAQMS-Aura $O_3$ is <10%.

Bias, correlation, and RMSE for each site are presented in Table 1. The SHADOZ stations within the Maritime Continent region are in bold font. These statistics are evaluated for all observations within 4 altitude ranges: surface- 5km, 5-10 km, 10-15 km, and 15-20km. The mean percent bias for the surface – 5km altitude range for all sites is 9.17%. The surface – 5km bias is larger than the mean at the Hilo, American Samoa, Costa Rica, San Cristobal, Nairobi, and Natal sites. This enhanced lower troposphere bias is associated with very low (< 20 ppbv) surface $O_3$ concentrations at American Samoa, San Cristobal, and

Hilo. RAQMS-Aura is moderately correlated (0.5-0.75) in time and space with SHADOZ between the surface and 5km for most sites. At the Kuala Lumpur site, RAQMS-Aura displays a small bias (6.909%) and a correlation of 0.458 with all SHADOZ ozone measurements. RAQMS-Aura strongly overestimates the surface $O_3$ concentration by >40% at Kuala Lumpur, though above the surface the average bias in this region is < 10% and the RAQMS-Aura $O_3$ analysis is moderately (0.5-0.8) correlated with SHADOZ. Between 5-10km, the mean percent bias is < ± 10% for all sites except Java where it is

20.22%. However, RAQMS-Aura has a correlation of 0.6585 with Java between 5 and 10km.

Overall, RAQMS-Aura does capture a substantial portion of the observed variability in tropical ozone profiles as indicated by the moderate to strong correlations with SHADOZ ozone profiles, though it does significantly overestimate near-surface ozone concentrations.

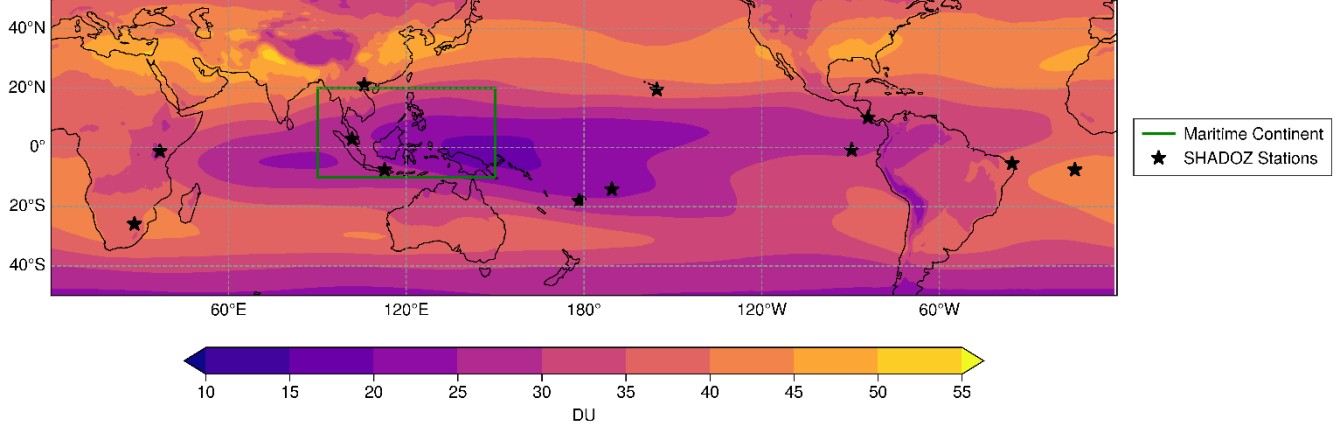

**Figure 8. SHADOZ ozonesonde sites (stars) and mean RAQMS-Aura tropospheric ozone column (contours).**

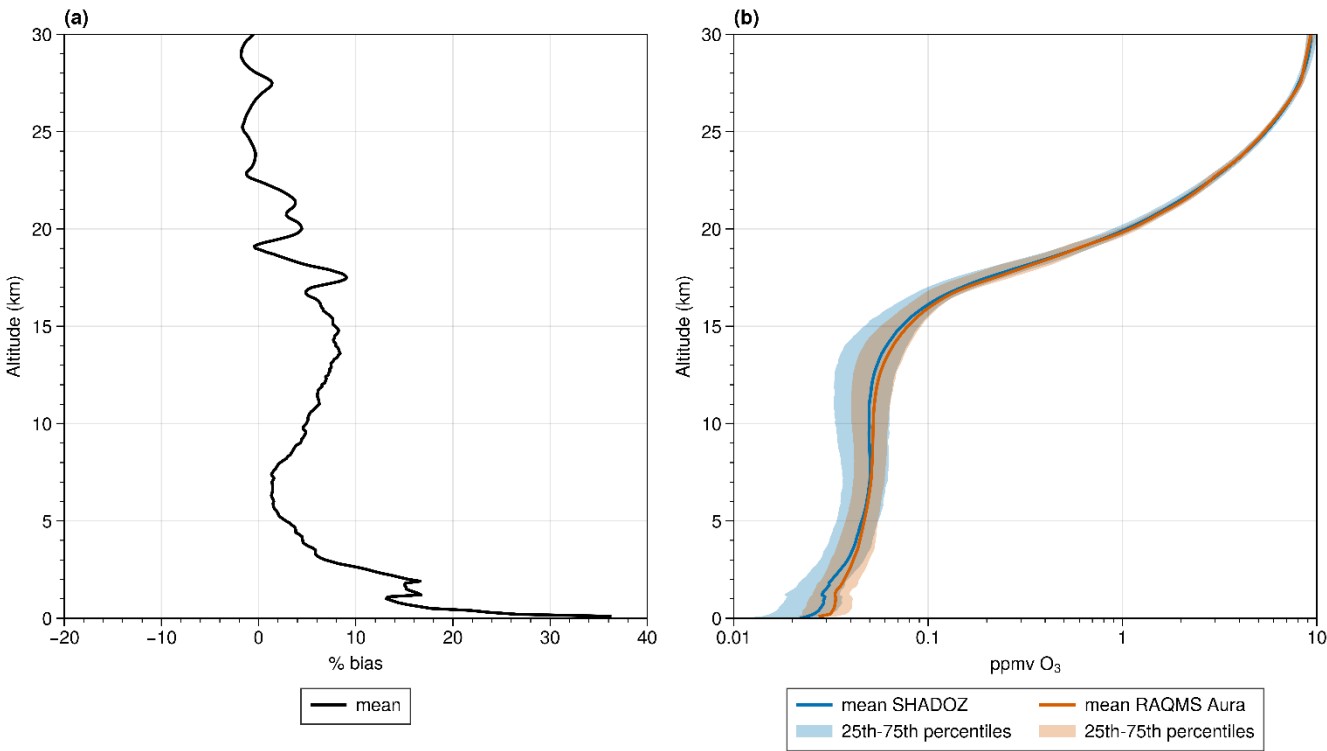

**Figure 9. Comparison of RAQMS-Aura O$_3$ mixing ratio to tropical SHADOZ ozonesondes. Panel a shows the percent bias in RAQMS-Aura relative to the ozonesondes. Panel b is percentiles for SHADOZ (blue) and RAQMS-Aura (orange).**


Table 1. Correlation, bias, and RMSE between SHADOZ ozonesondes and coincident RAQMS-Aura Ozone mixing ratio.

| | Number of profiles | Altitude Range | Correlation | RMSE (ppbv) | Mean Bias (ppbv) | Normalized Mean Bias (%) |
|---|---|---|---|---|---|---|
| American Samoa (14.2°S, 170.6°W) | 333 | 0-5 km | 0.7415 | 9.36 | 3.27 | 13.9 |
| | | 5-10 km | 0.6399 | 11.67 | 1.02 | 2.91 |
| | | 10-15 km | 0.6819 | 16.84 | 3.9 | 10.26 |
| | | 15-20 km | 0.9737 | 73.66 | -6.52 | -1.97 |
| Ascension Island (7.56°S, 14.22°W) | 237 | 0-5 km | 0.7675 | 13.29 | 2.54 | 5.66 |
| | | 5-10 km | 0.5743 | 14.07 | -0.76 | -1.18 |
| | | 10-15 km | 0.5799 | 17.13 | 7.08 | 11.16 |
| | | 15-20 km | 0.9654 | 67.15 | 11.17 | 4.00 |
| Costa Rica (10.0°N, 84.1°W) | 475 | 0-5 km | 0.5276 | 10.95 | 4.98 | 15.36 |
| | | 5-10 km | 0.3973 | 14.04 | 0.90 | 2.0 |
| | | 10-15 km | 0.4134 | 17.87 | 3.34 | 6.68 |
| | | 15-20 km | 0.9719 | 75.37 | 22.04 | 7.03 |
| Suva, Fiji (18.1°S, 178.4°E) | 135 | 0-5 km | 0.7828 | 9.53 | 1.86 | 6.7 |
| | | 5-10 km | 0.7517 | 12.02 | 0.81 | 1.93 |
| | | 10-15 km | 0.7907 | 15.28 | 7.15 | 17.9 |
| | | 15-20 km | 0.9712 | 84.02 | 6.49 | 1.83 |
| San Cristobal, Galapagos (0.92°S, 89.6°W) | 139 | 0-5 km | 0.7469 | 9.66 | 4.89 | 18.09 |
| | | 5-10 km | 0.5861 | 12.85 | 1.74 | 3.76 |
| | | 10-15 km | 0.5974 | 18.56 | 3.96 | 7.45 |
| | | 15-20 km | 0.9696 | 72.43 | -1.44 | -0.45 |
| **Hanoi, Vietnam (21.02°N, 105.8°E)** | 222 | 0-5 km | 0.7239 | 12.89 | -1.13 | -2.16 |
| | | 5-10 km | 0.6684 | 12.52 | 0.69 | 1.18 |
| | | 10-15 km | 0.7583 | 17.15 | 7.09 | 12.36 |
| | | 15-20 km | 0.9518 | 104.64 | 21.9 | 7.26 |
| Hilo, HI, USA (19.4°N, 155.4°W) | 534 | 0-5 km | 0.7464 | 12.32 | 5.96 | 15.68 |
| | | 5-10 km | 0.671 | 15.57 | 4.47 | 8.89 |
| | | 10-15 km | 0.8724 | 23.89 | 5.56 | 8.43 |
| | | 15-20 km | 0.9578 | 111.23 | 17.79 | 4.11 |

| | | | | | | |
|---|---|---|---|---|---|---|
| Irene, South Africa (25.9°S, 28.2°E) | 131 | 0-5 km | 0.6184 | 12.80 | -1.12 | -2.13 |
| | | 5-10 km | 0.7489 | 12.01 | -1.95 | -3.05 |
| | | 10-15 km | 0.8503 | 16.79 | -2.82 | -3.22 |
| | | 15-20 km | 0.9668 | 95.31 | 12.87 | 3.10 |
| **Watukosek, Java, Indonesia (7.6°S, 112.7°E)** | 104 | 0-5 km | 0.5556 | 13.62 | -1.94 | -5.2 |
| | | 5-10 km | 0.6585 | 13.39 | 7.02 | 20.22 |
| | | 10-15 km | 0.6911 | 16.54 | 12.09 | 40.91 |
| | | 15-20 km | 0.9602 | 82.66 | 27.41 | 10.44 |
| **Kuala Lumpur, Malaysia (2.73°N, 101.7°E)** | 197 | 0-5 km | 0.458 | 11.19 | 2.29 | 6.91 |
| | | 5-10 km | 0.5987 | 9.84 | 3.38 | 9.19 |
| | | 10-15 km | 0.5614 | 13.43 | 3.69 | 9.47 |
| | | 15-20 km | 0.9732 | 72.92 | 27.90 | 10.14 |
| Nairobi, Kenya (1.3°S, 36.8°E) | 447 | 0-5 km | 0.6276 | 9.84 | 3.98 | 10.74 |
| | | 5-10 km | 0.6438 | 13.89 | -0.17 | -0.33 |
| | | 10-15 km | 0.6543 | 17.61 | -0.92 | -1.53 |
| | | 15-20 km | 0.9758 | 63.95 | 11.34 | 3.78 |
| Natal, Brazil (5.4°S, 35.4°W) | 300 | 0-5 km | 0.8152 | 10.50 | 3.90 | 10.48 |
| | | 5-10 km | 0.7234 | 12.63 | -1.11 | -1.88 |
| | | 10-15 km | 0.7615 | 14.68 | 3.30 | 5.17 |
| | | 15-20 km | 0.9764 | 58.96 | -6.42 | -2.13 |
| All | 3254 | 0-5 km | 0.7712 | 11.32 | 3.33 | 9.19 |
| | | 5-10 km | 0.7221 | 13.38 | 1.29 | 2.61 |
| | | 10-15 km | 0.8103 | 18.13 | 3.89 | 7.02 |
| | | 15-20 km | 0.9666 | 82.35 | 11.92 | 3.61 |


## 3.3 ENSO Composites

Based on comparison of RAQMS-Aura total precipitation with TRMM 3B43 we conclude that RAQMS-Aura reasonably reproduces convection over the Pacific Ocean, particularly within the ITCZ. RAQMS-Aura captures the observed variability

in tropospheric ozone but has a ~2DU high bias relative to the OMI-MLS TOR. RAQMS-Aura captures the observed CO columns in the tropics very well. Based on comparison of RAQMS-Aura ozone profiles with SHADOZ profiles, we conclude that RAQMS-Aura reasonably captures observed variability in tropical ozone profiles but overestimates the near-surface concentrations. To characterize the anomaly associated with ENSO, composites for El Niño and La Niña periods are generated for precipitation, convective mass flux, diabatic heating, ozone concentration, carbon monoxide, and net ozone production
from monthly mean RAQMS-Aura analyses.

### 3.3.1 Precipitation

Composites of the de-seasonalized anomaly in precipitation for TRMM and RAQMS-Aura for positive ENSO and negative ENSO are given in figure 10. The TRMM and RAQMS-Aura composites are strongly correlated, with a spatial correlation of 0.77 for El Niño composites and 0.739 for the La Niña composites. The dominant feature of the El Niño phase in the TRMM
data and RAQMS-Aura re-analysis is an enhancement of precipitation in the tropics east from 150°E to the western coast of Central America and suppressed precipitation over the maritime continent. RAQMS-Aura however diverges from observations by displaying suppression of precipitation in regions around 7.5°S-39°S, 150°W-120°W and 7.5°N-20°N, 150°E-180°E where precipitation is enhanced in TRMM. During the La Niña phase, precipitation is suppressed over the central Pacific and enhanced over the maritime continent. For both TRMM and RAQMS-Aura the El Niño and La Niña composites are near
mirrors of one another, with the location of the maximum change shifted west during the negative phase from the positive phase.

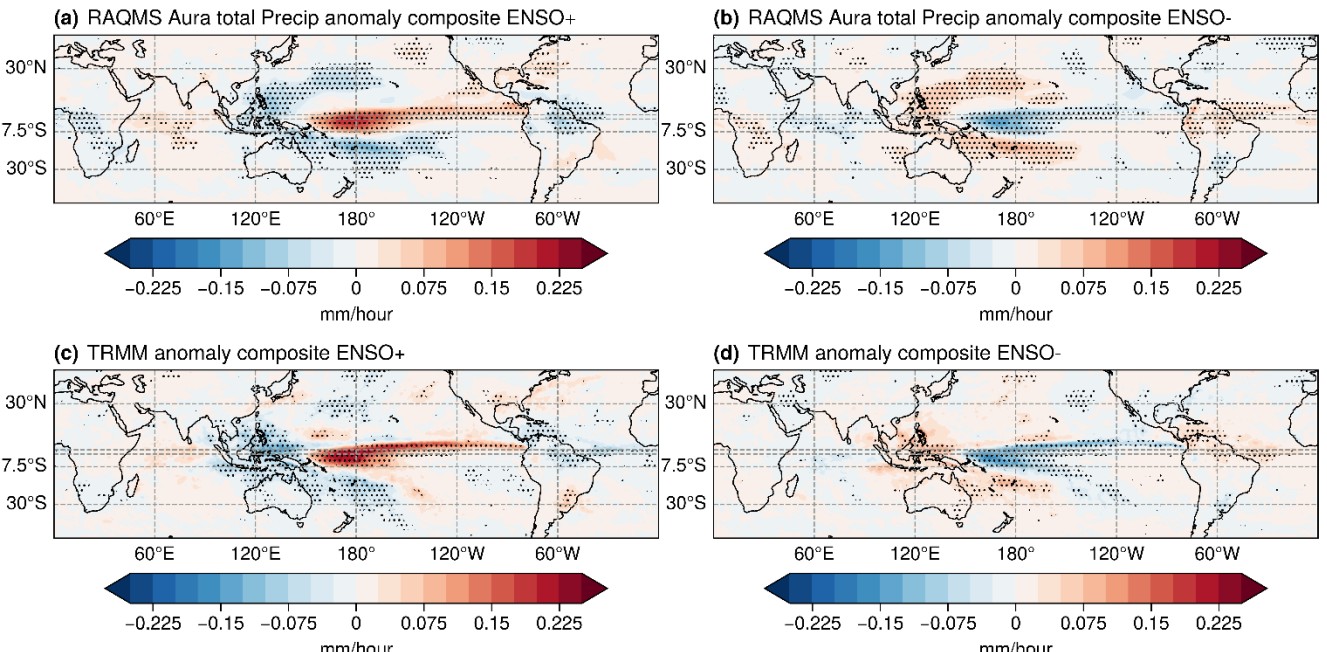

**Figure 10. Composited precipitation anomalies for El Niño in RAQMS-Aura (a) and TRMM 3B43 (b) and La Niña in RAQMS-Aura (c) and TRMM 3B43 (d). Shaded regions indicate where the composite is significant at the 95% confidence level from a t test.**

**3.3.2 Response of Tropospheric Total Column Ozone and Carbon Monoxide Column to ENSO**

ENSO composites for OMI-MLS TOR (Ziemke et al., 2006) and Measurements of Pollution in the Troposphere (MOPITT) CO (Emmons et al., 2004) are used to confirm the representativeness of RAQMS-Aura ENSO chemical signals.

Tropical tropospheric ozone column (TTOC) anomalies in RAQMS-Aura and the OMI-MLS TOR for the positive and negative phases of ENSO are shown in figure 11. TTOC anomalies are 1-2 DU larger during the positive phase of ENSO than in the negative phase. Within both the RAQMS-Aura TTOC and OMI-MLS TOR, El Niño is associated with an increase over the maritime continent and a decrease over the central and eastern Pacific Ocean. The decrease over the Pacific Ocean is flanked by increased concentrations to the north and south. Outside of the Pacific region, the tropospheric column anomaly associated with the ENSO phase is less than 1 DU. During La Niña, a small decrease in tropospheric ozone occurs over the maritime continent while an increase occurs over the central-eastern Pacific. The location of the peak decrease in TTOC in the eastern Pacific depicted in the El Niño composite is comparable to that found by Oman et al. 2011 and Olsen, Wargan, and Pawson 2016. Earlier studies of Peters et al., 2001, Doherty et al., 2006, and Ziemke and Chandra, 2003 show this peak decrease in TTOC is more towards the southeast. As our analysis is consistent with observations, the differences from earlier analyses are likely due to variability in ENSO and the influence of the large 2015 El Niño event during the 2006-2016 period under consideration in this study.

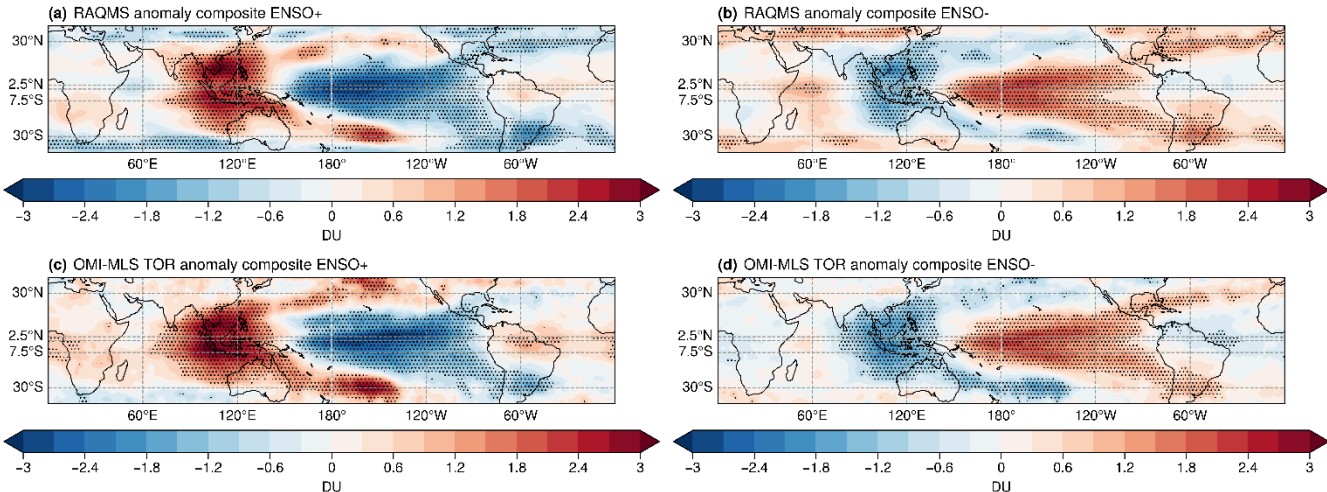

**Figure 11. Composited TTOC anomalies associated with El Niño in RAQMS-Aura (a) and OMI-MLS TOR (c) and La Niña in RAQMS-Aura(b) and OMI-MLS TOR(d). Shaded regions indicate where the composite is significant at the 95% confidence level from a t test.**

CO column anomalies for RAQMS-Aura and MOPITT are presented in figure 12. MOPITT CO anomalies appear nosier due to the sparse spatial sampling of the MOPITT instrument. RAQMS-Aura reproduces ENSO-related variability in CO as observed by MOPITT with both El Niño and La Niña composites having a spatial correlation of 0.850. RAQMS-Aura CO column is on average increased across the tropics during El Niño, with stronger enhancements of 0.4 x $10^{18}$ mol/cm$^2$ observed over the maritime continent. Enhanced CO over the maritime continent is tied to enhanced biomass burning during El Niño as precipitation is suppressed, increasing fuel aridity, and thereby increasing susceptibility to fire (Reid et al., 2013; van der Werf et al., 2017; Yin et al., 2016). RAQMS-Aura CO column decreases over the maritime continent during La Niña and is enhanced over South America. During La Niña, rainfall is enhanced over the maritime continent, resulting in CO decreases as fires are suppressed.

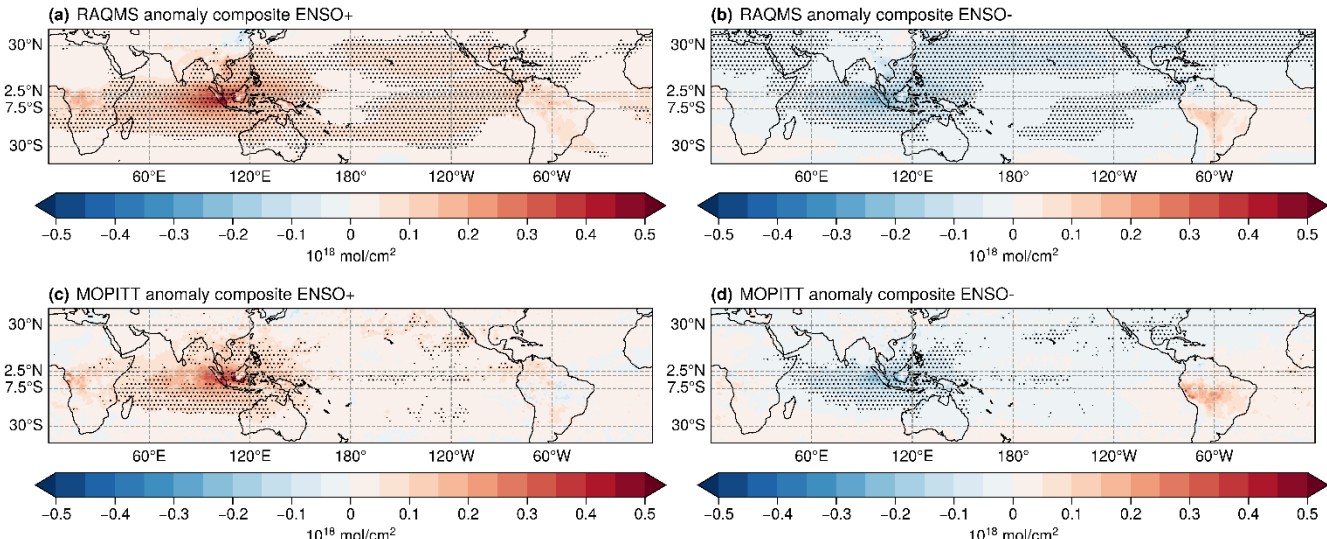

**Figure 12. Composited CO column anomalies associated with El Niño in RAQMS-Aura (a) and MOPITT (c) and La Niña in RAQMS-Aura(b) and MOPITT(d). Shaded regions indicate where the composite is significant at the 95% confidence level from a t test.**

### 3.3.3 Vertical structure of tropospheric response to ENSO

As this study utilizes reanalysis data, we can provide further context to the patterns in TTOC and CO columns. In particular, we explore how the vertical structure of convective mass flux, large-scale diabatic heating, and ozone production/loss terms respond to ENSO. Meridionally averaged vertical profile cross sections are calculated between 7.5°S and 2.5°N. This latitude band was selected as it cuts across the maximum and minimum precipitation anomalies associated with ENSO (fig 10) and for consistency with the cross-sections analyzed by Doherty et al. 2006.

Convective mass flux anomalies between 7.5°S and 2.5°N for the positive and negative phases of ENSO are presented in Figure 13. The strongest convective mass flux anomaly is over the Pacific Ocean during both the positive and negative phase of ENSO. This strong convective mass flux anomaly is also where the absolute maximum precipitation anomaly occurs, which is expected given the dominance in convective precipitation in this region. Diabatic heating anomalies presented in figure 14 are qualitatively similar to the convective mass flux ENSO anomalies. This is because the majority of the diabatic heating in this region is associated with the large-scale response to sub-grid-scale convective precipitation. The convective mass flux and diabatic heating anomalies during El Niño indicate decreased upward vertical transport over the maritime continent where precipitation is suppressed and increased upward vertical transport over the central Pacific where precipitation is enhanced. Conversely, the convective mass flux and diabatic heating anomalies during La Niña both indicate enhanced vertical transport over the maritime continent and increased downward vertical transport over the central Pacific. In Doherty et al. 2006 and Sudo and Takahashi 2001 the positive and negative mass flux anomalies are of similar magnitudes while here the negative flux anomaly over Micronesia is ½-⅓ the strength of the anomaly over the central-eastern Pacific. This may be a consequence

of the high bias in precipitation over Micronesia in the RAQMS-Aura reanalysis, as the precipitation anomaly El Niño composite indicates that precipitation is not suppressed as much as in observations over the region. However, these differences in the strength of the vertical motion anomalies are consistent with the ENSO precipitation anomaly over the central Pacific being larger than that of the anomaly over the maritime continent in TRMM observations and RAQMS-Aura analyses. The precipitation and mass flux anomaly patterns display suppressed (enhanced) vertical motion over the Pacific and enhanced

(suppressed) vertical motion over the maritime continent during the negative (positive) phase.

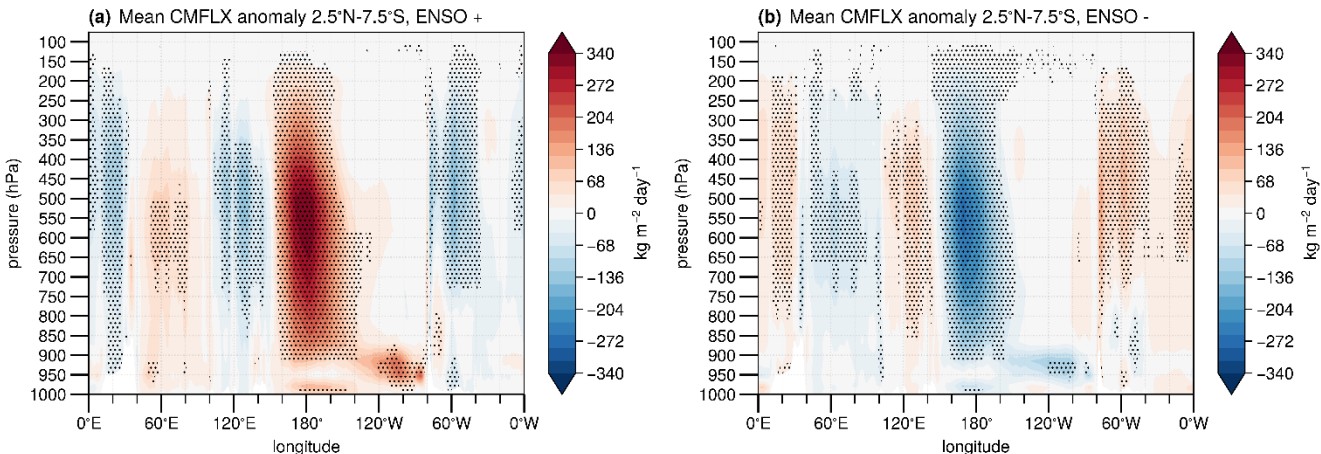

**Figure 93. RAQMS-Aura convective mass flux (CMFLX) anomalies for a) positive and b) negative ENSO phases. Shaded regions indicate where the composite is significant at the 95% confidence level from a t test.**

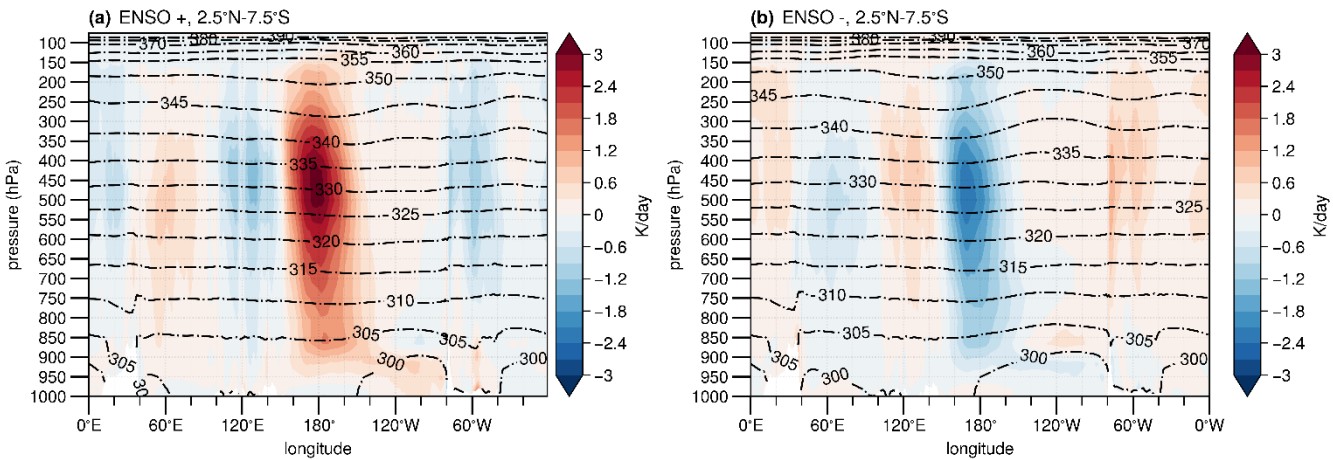

**Figure 104. RAQMS-Aura diabatic heating anomalies (colors) and theta (contours) for a) positive and b) negative ENSO phases.**

Ozone anomaly cross-sections associated with ENSO are presented in Figure 15. During El Niño the tropospheric ozone anomaly extends across the depth of the troposphere over the maritime continent, with two distinct stronger (>3 ppbv) enhancements above 550 hPa and below 700 hPa. Over the central Pacific (from 160°E to 140°W) where the convective mass flux is enhanced in the El Niño composite through the depth of the troposphere, a decrease in the ozone concentration of 3-5

ppbv occurs. The lower troposphere enhancement over the maritime continent is accompanied by a positive anomaly in net $O_3$ production (fig 17a), indicating that some of the enhancement in TTOC over the maritime continent during El Niño is due to enhancement in chemical production and not solely due to shifts in the circulation pattern. The El Niño ozone anomaly cross-section is <1 ppbv throughout the majority of the troposphere off the South American Coast, indicating that the TTOC decrease is due to the decreased (>9 ppbv) concentrations near the tropopause, above 200 hPa. The La Niña ozone anomaly cross-

section shows enhancement in ozone over the central Pacific and decrease over the maritime continent. Over the maritime continent a distinct stronger (>2 ppbv) decrease is seen below 700 hPa and above 350 hPa. Tropical upper troposphere ozone is also impacted by the quasi-biennial oscillation (QBO) (Oman et al., 2013). We evaluated the QBO signatures for both zonal mean zonal wind and ozone. We find RAQMS-Aura does a reasonable job of capturing the stratospheric QBO signal in both zonal mean zonal winds and ozone. However, we find the influence of the QBO on RAQMS-Aura ozone in the tropical upper

troposphere is smaller than the of ENSO influence during the 2006-2016 period considered in this study (Supplement).

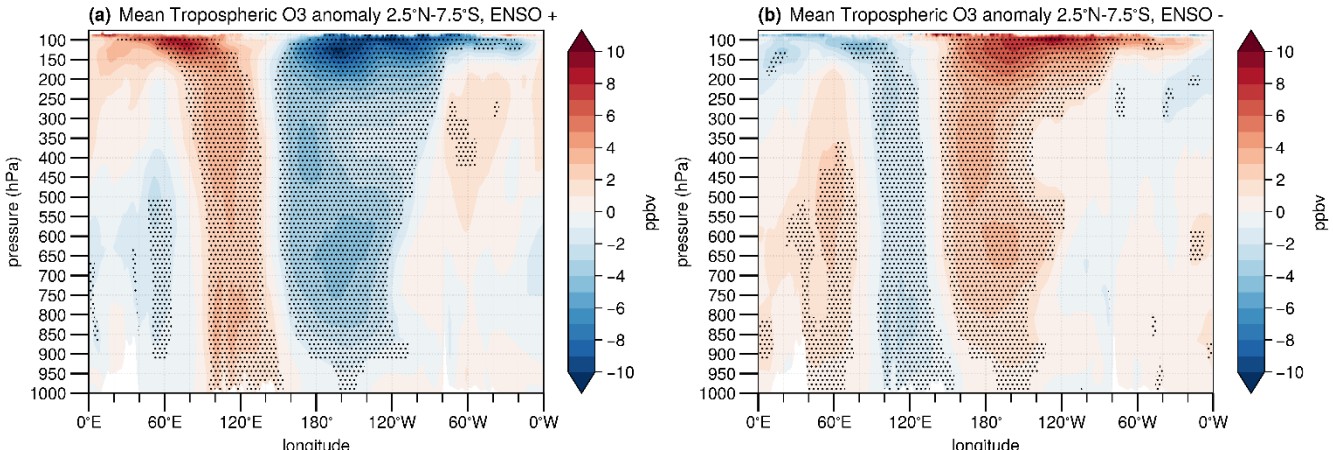

**Figure 115. Anomalies in RAQMS-Aura ozone profiles below the tropopause associated with a) El Niño and b) La Niña. Shaded regions indicate where the composite is significant at the 95% confidence level from a t test.**

CO anomaly cross-sections for each ENSO phase are presented in figure 16. Tropical CO is anomalously high during El Niño

and anomalously low during La Niña. Tropical CO is enhanced over the maritime continent during El Niño throughout the tropical troposphere, with the strongest enhancement near the surface indicative of a strong increase in biomass burning emissions. The near-surface enhancements in CO over South America and Africa during El Niño are also likely tied to CO emissions from biomass burning, though these enhancements are not spread through the depth of the troposphere as occurs over the maritime continent. The negative CO anomalies associated with La Niña are largest over the maritime continent and

are present through the depth of the troposphere. The enhancement in CO Column over South America associated with La Niña is not present in the La Niña vertical cross-section as it is to the south of the latitudes used to generate the cross-section composite.

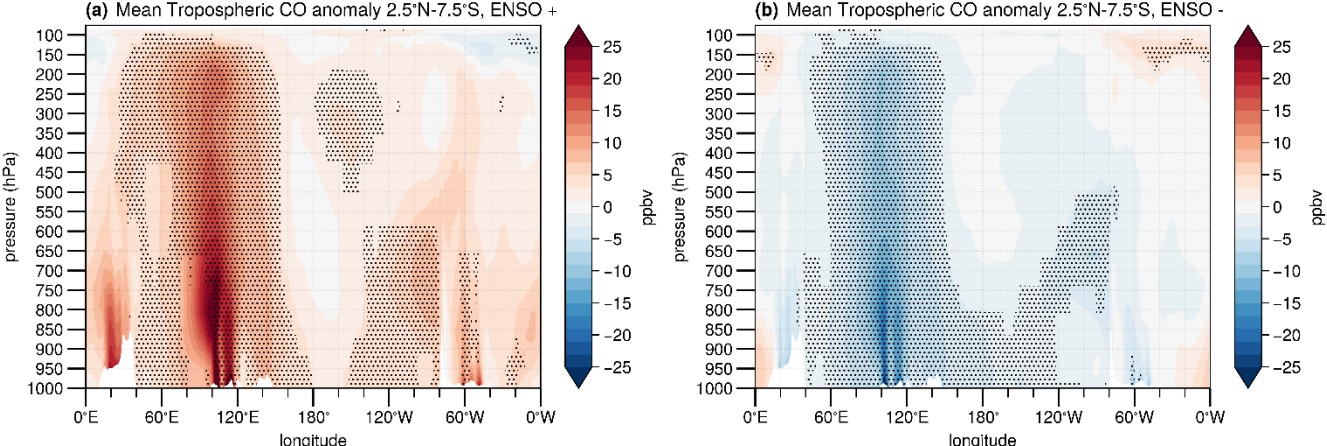

**Figure 126. Anomalies in RAQMS-Aura CO profiles below the tropopause associated with a) El Niño and b) La Niña. Shaded regions indicate where the composite is significant at the 95% confidence level from a t test.**

Net ozone production (production - loss terms) anomalies are presented in Figure 17. RAQMS has standard hydrogen oxides (HO$_x$), chlorine oxides (ClO$_x$), bromine oxides (BrO$_x$), and NO$_x$ ozone photochemistry (Eckman et al., 1995) with Carbon Bond-Z (CB-Z) (Zaveri and Peters, 1999) treatment of non-methane hydrocarbon chemistry. Chemical production and loss are calculated explicitly for the O$_x$ family, which in RAQMS includes O($^1$D), O($^3$P), O$_3$, NO$_2$, HNO$_3$, NO$_3$, N$_2$O$_5$, HNO$_4$, PAN (peroxynitrates), and MPAN. Since the shifts in precipitation within the tropics are largely associated with shifts in convective clouds (fig. 1) and the photolysis rates in RAQMS respond only to changes in atmospheric transmittance due to large-scale resolved clouds, changes in net ozone production associated with changes in convective cloud distributions are not accounted for in this study. The largest net ozone production anomalies are closest to the surface and below 700 hPa. The change in net ozone production is smaller in La Niña than El Niño. Enhanced production of 2-3 ppbv/day is found over central Africa, Indonesia, and the Amazon rainforest in Brazil. These regions show reductions of ~ 1.3 ppbv/day in ozone production in the La Niña composite. El Niño is known to increase fire emissions in Indonesia as a consequence of the decreased rainfall over the region (Field et al., 2016; Park et al., 2021), and so the increased production of ozone during El Niño captured by RAQMS-Aura is likely to be partially due to enhanced chemical production of ozone in biomass burning plumes. Enhanced production during El Niño occurs over all 3 biomass burning regions but only the maritime continent shows a significant (>4 ppbv) enhancement in O$_3$ below 700 hPa. In contrast, the enhanced production over South America and Africa is associated with weak (<2 ppbv) ozone enhancement. The average winds below 800hPa during El Niño over South America (not shown) are northeasterly, resulting in transport of the ozone associated with biomass burning to the south and out of the latitudes included in the cross-section (7.5°S to 2.5°N). Over the maritime continent, the average winds below 750 hPa are southerly and decline in strength through the cross-section. Based on these wind patterns, ozone associated with biomass burning over the maritime continent experiences less meridional transport and has stronger influences on the ozone profile within this meridional cross-section.

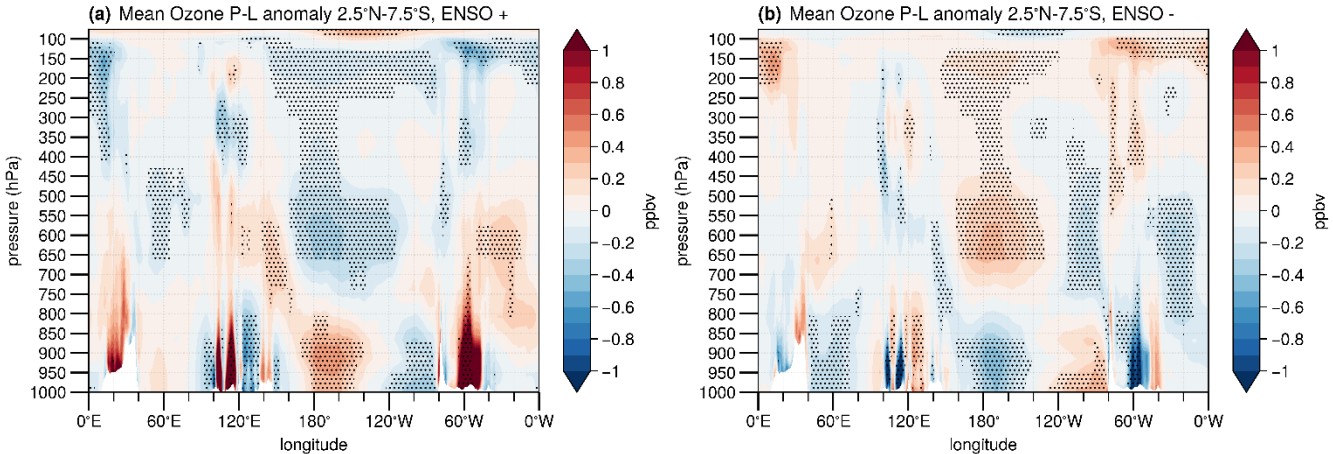

**Figure 137. Anomalies in RAQMS-Aura net O$_3$ production associated with a) El Niño and b) La Niña. Shaded regions indicate where the composite is significant at the 95% confidence level from a t test.**

## 3.4 EOF Analysis

In addition to composite analysis, Empirical Orthogonal Function (EOF) analysis is used to investigate the role played by ENSO in TTOC variability. The first EOF of TTOC has been previously found to be associated with ENSO, while TTOC EOFs 2 and 3 are uncorrelated with ENSO (Doherty et al., 2006; Sekiya and Sudo, 2012). ENSO positive and negative phases are near opposites of each other, and so it is reasonable to expect that much of the variability associated with ENSO can be captured with a single EOF. The EOF spatial patterns are displayed for TTOC, precipitation, and CO column in figures 18, 19, and 20. PC time series are presented in figure 21, alongside the Niño 3.4 index for reference.

### 3.4.1 EOFs

EOF patterns for TTOC are displayed in figure 18. The TTOC PC$_1$ has a temporal correlation of 0.747 with the Niño 3.4. The associated EOF indicates a 2-2.5 DU enhancement over the maritime continent and a 1.6-2 DU decrease over the Pacific (figure 18a). EOF$_1$ captures similar features to those in the El Niño TTOC composite, though the enhancement in TTOC near Vietnam is weaker relative to the enhancement near Indonesia in the EOF compared to the composite. TTOC PC$_2$ and PC$_3$ are weakly correlated with the Niño 3.4 index, with temporal correlations of -0.144 and -0.209 respectively. TTOC EOF$_2$ explains around half as much variance as TTOC EOF$_1$ and shows a wave 1 like pattern with a peak in the northeast Pacific. TTOC EOF$_3$ accounts for changes of less than 1 DU on average, and a maximum near 3 DU. At the most, this is ~10% of the mean TTOC and less than 1% on average. TTOC EOF$_3$ captures an increase across the equatorial Pacific and decreases elsewhere.

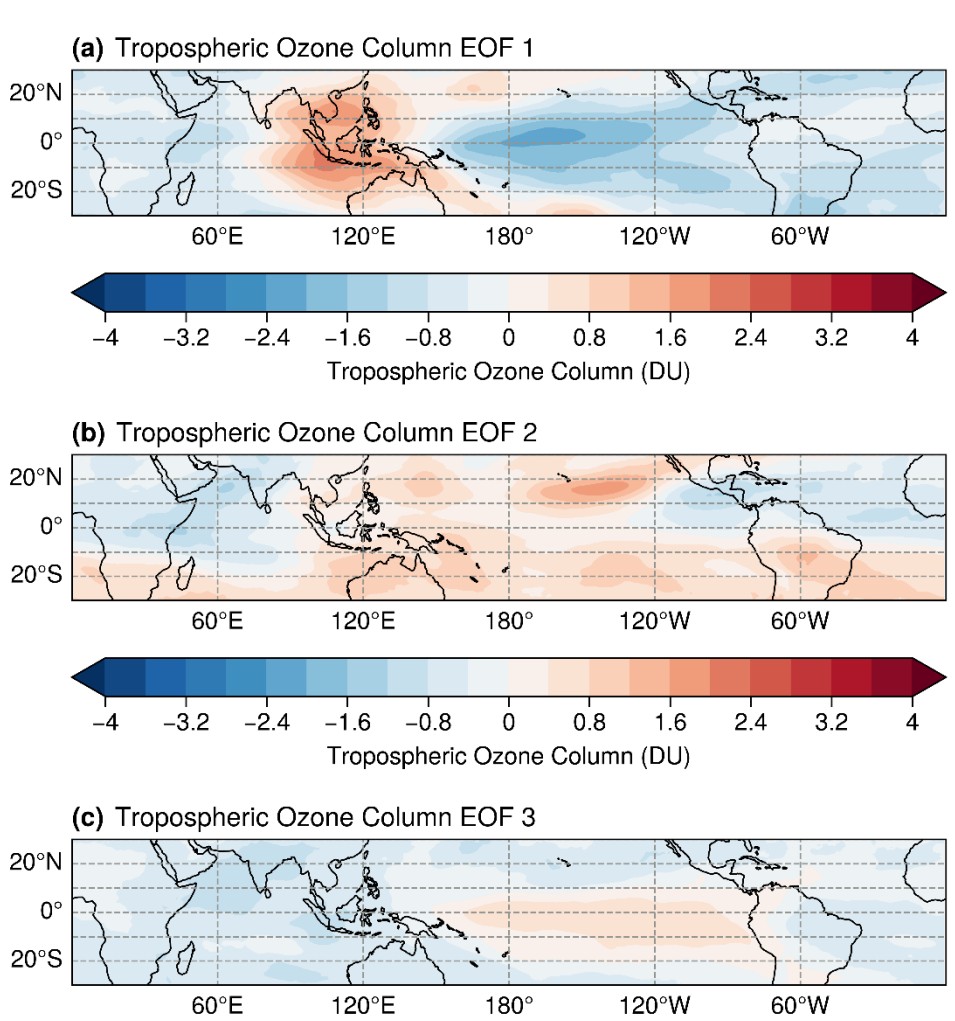

**Figure 148. Patterns for RAQMS-Aura TTOC EOF 1-3, scaled by 1 standard deviation of the associated PC. EOF$_1$ explains 17.20% of the non-seasonal variance in TTOC, EOF$_2$ explains 8.70% and EOF$_3$ explains 6.00%.**

EOF patterns for total precipitation are displayed in figure 19. The precipitation PC$_1$ is strongly correlated with the Niño 3.4

index, with a temporal correlation of 0.870, as well as a strong temporal correlation with the TTOC PC$_1$ (0.818). The associated

EOF pattern is similar to the El Niño precipitation composite in figure 10a, though the magnitude of the decreased precipitation

in the western Pacific relative to the enhancement in the central Pacific is smaller than in the composite. Precipitation EOFs 2

and 3 combined capture a similar amount of variability in precipitation as EOF$_1$ alone. Their PCs are not correlated with the

Niño 3.4 index, with a PC$_2$ temporal correlation of -0.02, and a PC$_3$ temporal correlation of -0.093. The EOF$_2$ pattern depicts

a small, localized enhancement in the central southern Pacific Ocean, slightly stronger enhancements of ~0.06 mm/hour in the

Caribbean and NW equatorial Pacific, and decreased precipitation in the remainder of the northern hemisphere Pacific. The EOF$_3$ pattern accounts for changes of <0.03 mm/hour on average. The largest of these small changes are a decrease in precipitation in the central Pacific to the east of where the maximum precipitation anomaly associated with ENSO is located. Precipitation PC$_3$ has a correlation of 0.695 with TTOC PC$_2$, indicating there is some co-variability between the two.

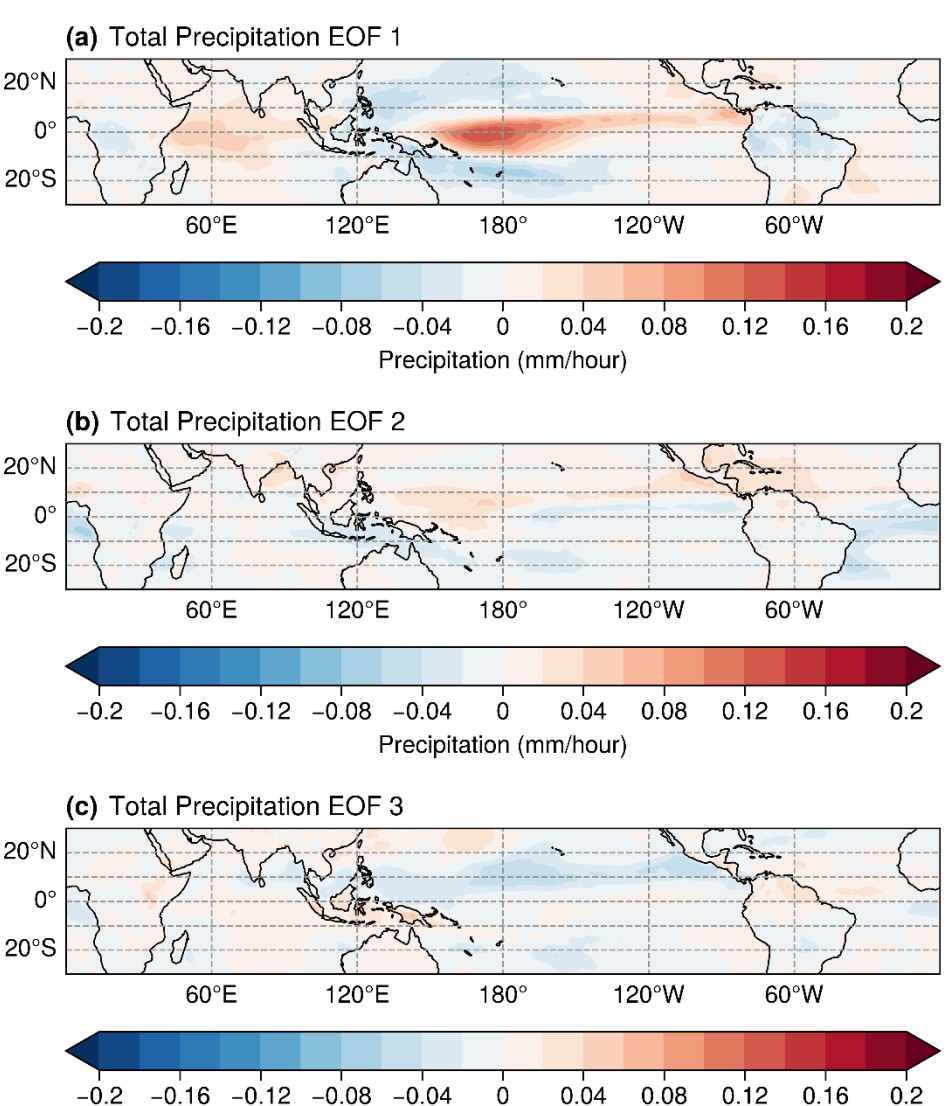

**Figure 159. Patterns for RAQMS-Aura total precipitation EOF 1-3, scaled by 1 standard deviation of the associated PC. EOF1 explains 8.33% of the non-seasonal variance in total precipitation, EOF2 explains 4.73% and EOF3 explains 4.46%.**

EOF patterns for CO column are displayed in figure 20. Inter-annual variability in tropical CO has been shown to be predominately influenced by biomass burning emissions (Rowlinson et al., 2019). All 3 CO column EOF patterns appear to be heavily influenced by extreme biomass burning events, as the strongest changes are over the maritime continent and South

America and the peaks in the PCs correspond with years with enhanced biomass burning in the regions highlighted by the largest values in the EOF (eg. van der Werf et al., 2017). CO PC amplitude peaks are larger than 2 for $PC_1$ in late 2015; $PC_2$ in 2006, 2007, 2010, and 2015; and $PC_3$ in 2006, 2015, and 2016 (figure 21). $EOF_1$ explains 46.96% of the non-seasonal variance in CO, while $EOF_2$ explains 9.46% and $EOF_3$ explains 6.48%.

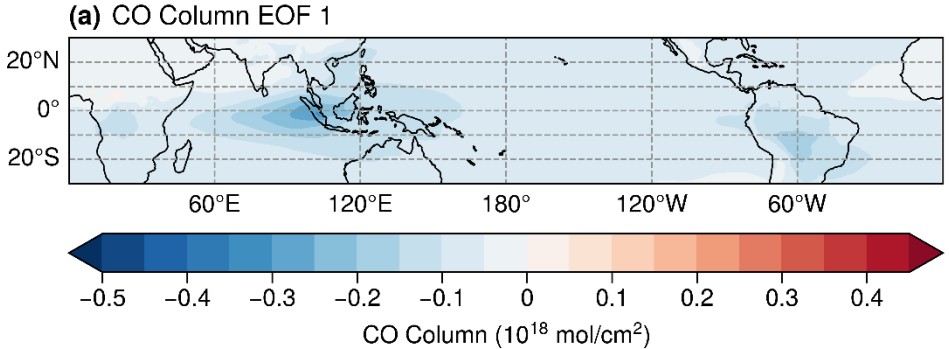

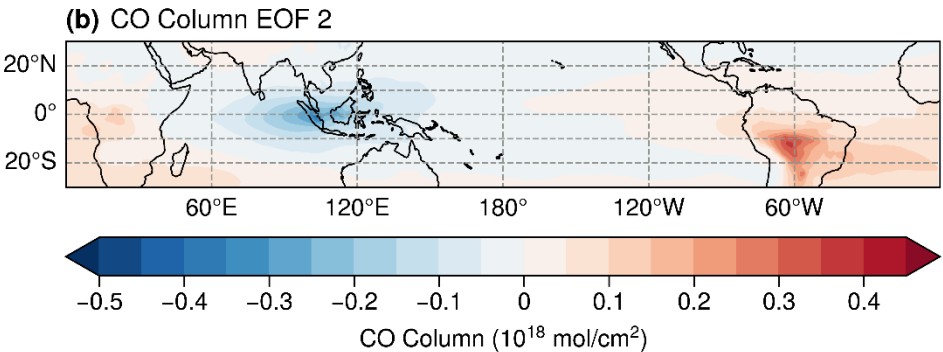

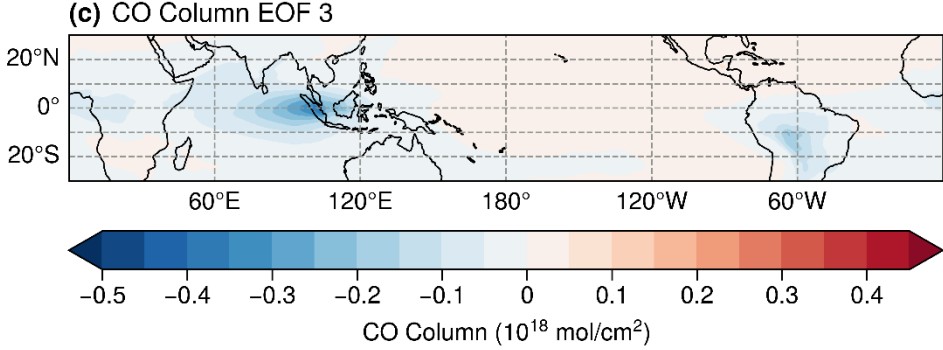

**Figure 20. Patterns for RAQMS-Aura CO Column EOF 1-3, scaled by 1 standard deviation of the associated PC. $EOF_1$ explains 46.96% of the non-seasonal variance in CO column, $EOF_2$ explains 9.46% and $EOF_3$ explains 6.48%.**

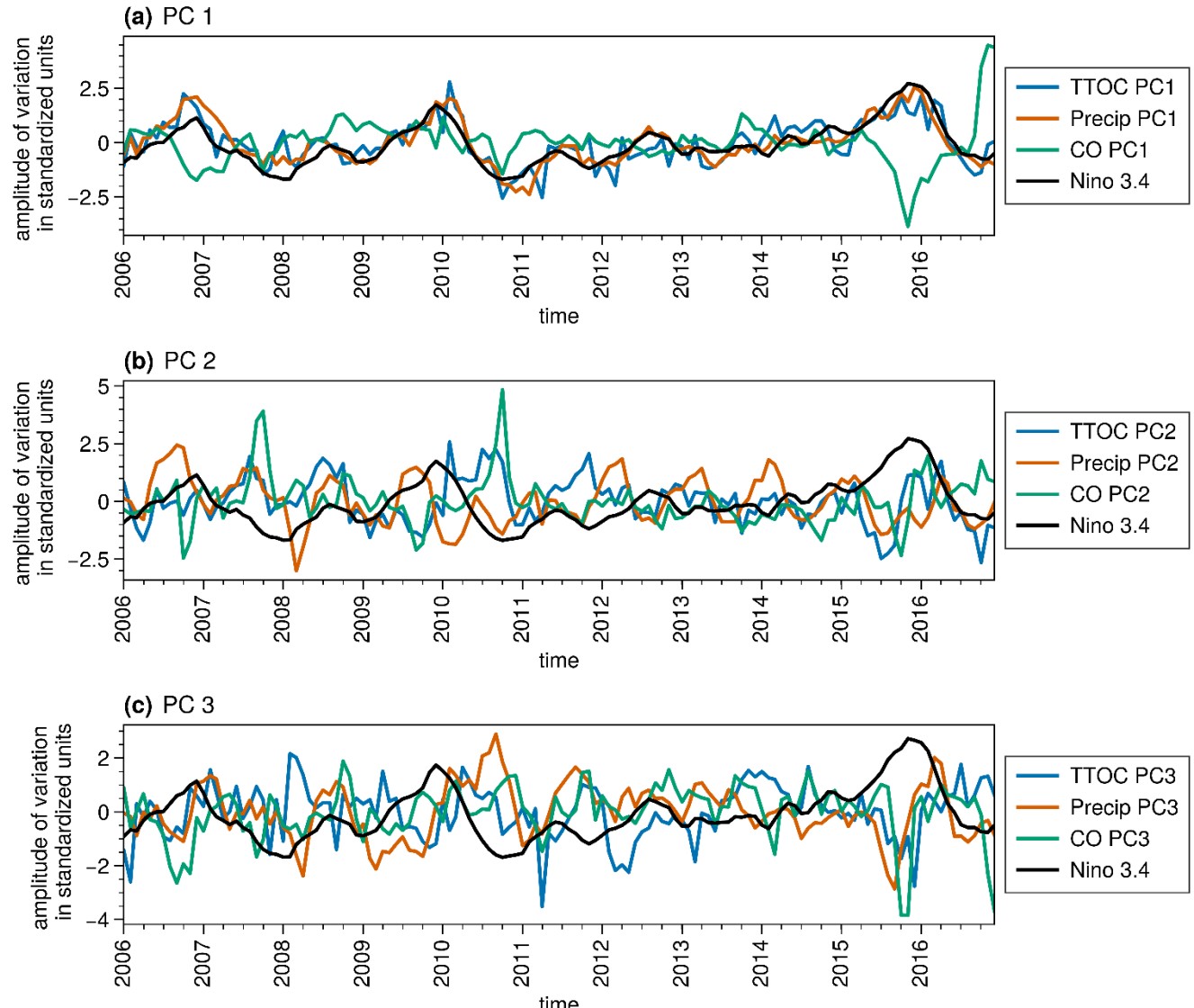

**Figure 21. Timeseries of PC$_1$ (a), PC$_2$ (b), and PC$_3$ (c) for TTOC, total precipitation, and CO Column. Niño 3.4 Index time series included for reference.**

Most variability in CO columns from 2006-2016 is explained by EOF$_1$. The physical pattern is indicative of a tropics-wide decrease (increase) in CO, with the peak change of ~0.3 x 10$^{18}$ mol/cm$^2$ centered over the maritime continent. CO PC$_1$ has a temporal correlation of -0.399 with the Niño 3.4 index, which indicates an ENSO influence on CO variability. Additionally, CO PC$_1$ is temporally correlated with precipitation PC$_1$ (-0.435), suggesting that ENSO related changes in precipitation contribute to the ENSO driven CO variability. This is consistent with precipitation influences on biomass burning. The CO EOF$_2$ pattern shows CO column enhancements over Brazil and decreases over the maritime continent. CO PC$_2$ has a temporal correlation of -0.297 with the Niño 3.4 index, and temporal correlation of -0.435 with TTOC PC$_1$, suggesting that ENSO

related changes in CO contribute to ENSO driven TTOC variability. EOF$_3$ pattern again highlights the maritime continent and Brazil varying together, with an opposing change in CO across the Pacific. CO PC$_3$ displays a correlation of -0.145 with the Niño 3.4 index.

### 3.4.2 Multiple Linear Regression reconstruction of TTOC PC$_1$

From the composite analyses we are able to show that ENSO related shifts in precipitation correspond with changes in vertical motion, CO concentration, net ozone production, and tropospheric ozone concentrations. The composite analysis also indicates that some of the enhancement in TTOC over the maritime continent during El Niño is due to enhanced production of ozone from biomass burning emissions. The EOF analysis further links variation in biomass burning to the TTOC variation as CO PCs 1 and 2 are mildly temporally anti-correlated with TTOC and precipitation PC$_1$. This negative correlation is due to the suppression of biomass burning during precipitation. To quantify the relative importance of dynamical and biomass burning variability on ENSO related variability in TTOC, a multiple linear regression analysis is constructed using the principal components. The regression equation is shown in equation (1).

$$PC1_{TTOC} = w_1 PC1_{CO} + w_2 PC2_{CO} + w_3 PC3_{CO} + w_4 PC1_{precip} + e \quad (1)$$

The principal components are from the EOF analysis; $w_1$, $w_2$, $w_3$, $w_4$, and e are regression coefficients as determined using a least squares fit. The resulting regression model is shown in equation (2).

$$PC1_{TTOC} = 0.11* PC1_{CO} - 0.2 * PC2_{CO} + 0.004*PC3_{CO} + 0.8 * PC1_{precip} - 3.3x10^{-10} \quad (2)$$

This multiple PC regression reproduces the PC1$_{TTOC}$ very well, with the regression-based estimate correlating with the original PC1$_{TTOC}$ at 0.85 (fig 22a).

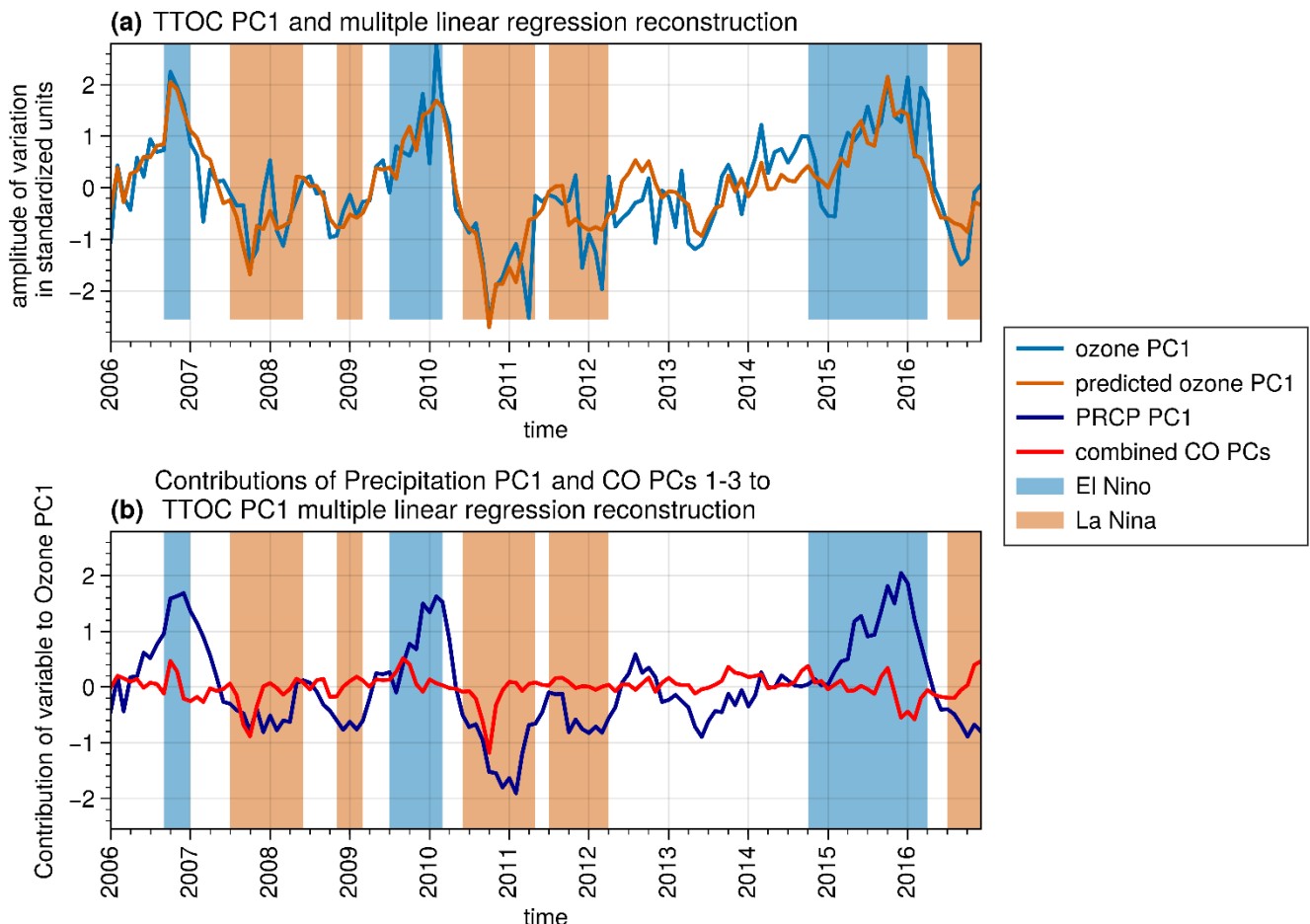

**Figure 22. a) TTOC PC$_1$ from EOF analysis and reconstructed from multiple linear regression. b) Contribution to regression of Precipitation PC$_1$ and combined contribution of CO PCs 1-3.**

The strongest weighted PC in the regression is the precipitation PC$_1$, which is expected given its strong correlation with TTOC
PC$_1$. This supports the result from Doherty et al. 2006 and Inness et al. 2015 that ENSO variability in TTOC is primarily driven by convective transport. The weights for CO PC$_1$ and PC$_2$ are also significant, indicating that CO, as a proxy for biomass burning, also contributes to TTOC variability.

A timeseries showing the contributions of precipitation PC$_1$ and the combined CO PCs to the TTOC PC$_1$ predicted by the regression is shown in figure 22b. The precipitation PC$_1$ regression contribution is positive during El Niño periods and negative
during La Niña periods. The combined regression contribution of the CO PCs shows that variability in CO contributes to ENSO variability in TTOC in an episodic way. As the CO column anomaly is linked to anomalous biomass burning emissions and net ozone production near the surface, it can be concluded that a portion of the ENSO variability in TTOC is due to biomass burning though it is a smaller portion than that linked to the dynamical effects of ENSO.

Additionally, each component of the regression can be removed independently in order to evaluate the impact of co-variability

between the CO PCs and precipitation $PC_1$ on the overall fit. RMSE and $R^2$ for the standard fit and the alternate fits are given in table 2. $R^2$ is maximized and RMSE minimized for the case where all CO PCs are considered. The poorest fit is obtained when precipitation $PC_1$ is removed. The linear regression that relates ENSO TTOC variability to only ENSO precipitation variability performs similarly to the regression with CO $PC_2$ removed, highlighting that the redistribution of $O_3$ and $O_3$ precursors by convection is the most significant contributor to ENSO variability in TTOC. The best regression fits ($R^2 > 0.7$)

include CO $PC_2$ and precipitation $PC_1$. This confirms that while variability in CO is not independent of precipitation, it does meaningfully contribute to ENSO variability in TTOC.

Table 2. RMSE and $R^2$ for TTOC $PC_1$ multiple linear regression models.

| Regression equation | $R^2$ | RMSE |
|---|---|---|
| $PC1_{TTOC} = 0.11* PC1_{CO} - 0.2 * PC2_{CO} + 0.004*PC3_{CO} + 0.8 * PC1_{precip} - 3.3x10^{-10}$ | 0.724 | 0.5258 |
| $PC1_{TTOC} = -0.2177*PC2_{CO} - 0.0526*PC3_{CO} + 0.7440 * PC1_{precip} - 2.072x10^{-10}$ | 0.714 | 0.5347 |
| $PC1_{TTOC} = 0.1433*PC1_{CO} - 0.0262*PC3_{CO} + 0.8752 * PC1_{precip} - 4.507x10^{-10}$ | 0.687 | 0.5591 |
| $PC1_{TTOC} = 0.1151*PC1_{CO} - 0.1984*PC2_{CO} + 0.8102 * PC1_{precip} - 5.293x10^{-10}$ | 0.722 | 0.5273 |
| $PC1_{TTOC} = -0.2373*PC1_{CO} - 0.4351*PC2_{CO} - 0.2023*PC3_{CO} + 9.887x10^{-10}$ | 0.287 | 0.8446 |
| $PC1_{TTOC} = 0.812 * PC1_{precip} - 4.777x10^{-10}$ | 0.669 | 0.5750 |

As inferred from the regression, El Niño increases in TTOC over the maritime continent are associated with CO $PC_1$ CO

enhancements over the maritime continent while CO $PC_2$ is associated with enhancements in CO over South America and Africa and decreases over Indonesia. Timeseries of the CO column and TTOC anomalies (not shown) have a temporal correlation of 0.668 over the maritime continent and 0.566 over South America. The TTOC and CO anomalies over the maritime continent are positive during El Niño events and negative during La Niña events. Over South America, the sign of the TTOC and CO anomalies are less consistent with ENSO phase.

**3.5 2015/2016 extreme El Niño**

Through the satellite era, extreme El Niño events in 1982/1983, 1997/1998, and 2015/2016 have been observed alongside weak and moderate events. These extreme events have a larger impact on the distribution of TTOC and have a larger contribution from biomass burning emissions than weaker El Niño events (Doherty et al., 2006; Inness et al., 2015). The 2015/2016 extreme El Niño was the strongest El Niño since the 1997/1998 event (Santoso et al., 2017). 2015 and 1997 are

also among the most extreme maritime continent biomass burning events, with 1997 ranking first followed by 2015 in an analysis of surface visibility at airports in Sumatra and Kalimantan from 1990-2015 (Field et al., 2016). Here we investigate how the inclusion of the 2015 extreme El Niño influences our interpretation of the importance of biomass burning on TTOC ENSO variability. As in prior analyses (Chandra et al., 1998, 2009; Sudo and Takahashi, 2001), we focus on October as biomass burning in the maritime continent peaks around October and would have its greatest impact on TTOC around the

same time (Field et al., 2016). In RAQMS-Aura, the CO PC amplitudes have the largest variability in October and the largest contributions of the CO PCs to the TTOC $PC_1$ regression occur in October.

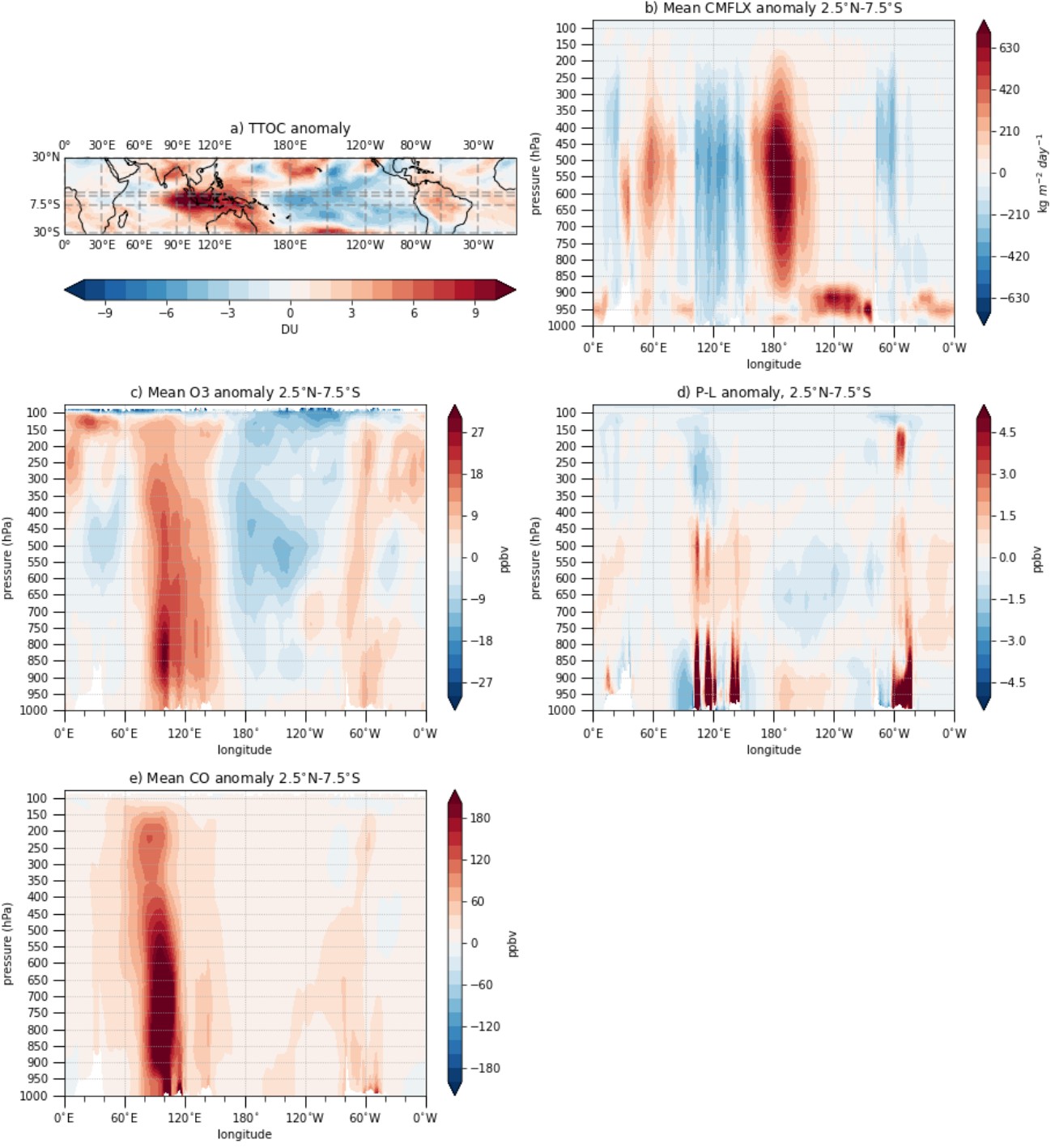

**Figure 23. RAQMS-Aura October 2015 a) TTOC anomaly, b) convective mass flux anomaly, and c) tropospheric ozone profile anomaly, d) P-L, e) CO.**

The RAQMS-Aura 2015 October TTOC anomaly is shown in figure 23a. This pattern is similar to the October 1997 anomaly in TTOC modeled by Sudo and Takahashi 2001 with an increase over the maritime continent that is 2-3 times stronger than the decrease over the eastern Pacific. However, the peak decrease over the eastern Pacific is more towards the central Pacific during 2015 than in 1997. The maximum increase over the maritime continent is 10-15 DU in October 2015, less than the maximum 20-24 DU increase in October 1997. RAQMS-Aura TTOC increases over South America in October 2015 by 1-4

DU, while the Sudo and Takahashi simulated October 1997 changes by less than 2 DU over South America. These differences over Africa and South America in 2015 versus 1997 are consistent the differences in patterns of convective mass flux. In 2015 mass flux is decreased aloft over Brazil and Africa (fig 23b), while in 1997 changes in mass flux over Brazil and Africa are weaker and are slightly positive (Sudo and Takanashi, 2001). The core of the upward mass flux anomaly over the Pacific is ~30-40 degrees closer to the dateline in 2015.

Over the maritime continent, the ozone concentration anomaly below 650 hPa is stronger than in the 2006-2016 El Niño average (fig 23c). This is linked to stronger ozone production in October 2015 (fig 23d). This enhancement in $O_3$ production in 2015 is likely due to increased fire activity, as CO column is increased throughout the tropics in 2015 (fig 23e) and the CO anomaly over the maritime continent is more widespread and stronger by $\sim0.2 \times 10^{18}$ mol/cm$^2$ than the 2006-2016 El Niño average. There is also an enhancement in CO, ozone, and net ozone production over South America in October 2015 relative

to the 2006-2016 El Niño composite. This shows that the biomass burning activity in 2015 was anomalous compared to the other El Niño years included in the RAQMS-Aura reanalysis, with significant burning occurring over both South America and the maritime continent.

**4 Conclusions**

    The RAQMS-Aura reanalysis captures observed ENSO variability in TTOC, CO, and precipitation. ENSO composites of

tropospheric ozone, carbon monoxide, convective mass flux, diabatic heating, and ozone net chemical production show that the observed ENSO signatures in TTOC result from a combination of convective redistribution and variability in production of ozone from biomass burning emissions, which are modulated by ENSO variability in precipitation. The location of the peak decrease in TTOC resulting from increased vertical motion in the eastern Pacific depicted in the El Niño composite found by this study is comparable to other studies of TTOC variability in the 2000s and 2010s (Olsen et al., 2016; Oman et al., 2011).

The location of the peak decrease in TTOC contrasts with that found by analyses of 1970s-2000 where it is more towards the southeast and near the South American coast (Doherty et al., 2006; Peters et al., 2001; Ziemke and Chandra, 2003). The RAQMS-Aura El Niño TTOC composite is in agreement with the El Niño composite OMI-MLS TOR observations, and the analysis of convective flux indicates that the ozone decreases over the central Pacific are due to enhanced vertical motion. Therefore, we believe the difference in position of the peak decrease in TTOC is due to characteristics of El Niño during our

analysis period. El Niño events from 2006-2016 were predominately El Niño Modoki events, while El Niño events between 1979 and 2002 display greater variability in type of El Niño and includes more canonical ENSO events (Hou et al., 2016; Lee

and McPhaden, 2010; Santoso et al., 2017). The ascending branches of Walker circulation cell is over the central Pacific during El Niño Modoki (Ashok et al., 2007), while during canonical El Niño the ascending branch is over the eastern Pacific. Since TTOC is decreased where vertical motion is enhanced during ENSO and increased where vertical motion is suppressed, it is expected that under El Niño Modoki conditions the largest decrease in TTOC will be in the central Pacific with TTOC increases in the western and eastern Pacific. This response of TTOC to El Niño Modoki is shown by Hou et al. 2016 and is in-line with the El Niño RAQMS-Aura TTOC anomaly composite calculated by this study (Fig 11a).

The strongest ENSO variability in tropospheric ozone is shown to occur near the tropopause. An enhancement in ozone below 700 hPa during El Niño occurs over the maritime continent that is dependent on the magnitude of the biomass burning emissions. The EOF analyses and multiple linear regression further indicate that ENSO variability in TTOC is driven by shifts in the location of the ascending and descending branches of the Walker circulation. The EOF and multiple linear regression analyses also indicate that variability in biomass burning, as inferred from CO anomalies, contributes to ENSO variability in TTOC. During the 2015/2016 strong El Niño event TTOC, CO, and convective mass flux anomalies were stronger than in the weaker ENSO events captured by the RAQMS-Aura reanalysis. The 2015 CO concentrations align with the mode captured by CO $EOF_1$ while the other El Niño years in our analysis align with the mode in CO $EOF_2$. Biomass burning enhanced TTOC and CO anomalies occurred over both South America and the maritime continent in October 2015 in contrast to the other El Niño years between 2006 and 2016 where biomass burning enhanced TTOC and CO was only found over the maritime continent.

### Code and data availability

The software used to generate the figures in this work and all raw data can be provided upon request by the corresponding author.

### Author Contributions

Bruckner and Pierce planned the study and collaborated on the analysis. Lenzen performed the RAQMS-Aura reanalysis. Bruckner wrote the paper. Pierce reviewed and edited the paper.

### Competing Interests

The contact author has declared that none of the authors has any competing interests.

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
