# Peer review of "Examining ENSO related variability in tropical tropospheric ozone in the RAOMS-Aura chemical reanalysis"

_EGUsphere, 2024_

## Author Comment (AC1)

Title: Examining ENSO related variability in tropical tropospheric ozone in the RAQMS-Aura chemical reanalysis

Author(s): Maggie Bruckner et al.

MS No.: egusphere-2024-1178

MS type: Research article

We appreciate the comments made by both reviewers and have revised the manuscript to address the specific comments and suggestions. This includes a new section that discusses comparison of the reanalysis with SHADOZ ozonesondes, OMI-MLS TOR, and MOPITT CO column and a supplemental discussing the influence of the QBO on upper tropical tropospheric ozone. We feel that this strengthens the manuscript by providing further detail regarding the fidelity of the reanalysis and the magnitude of the ENSO signature in tropospheric ozone relative to other influences.

The following is the reviewer's comments and our response to each comment in blue.

**Response to Referee #1**

Bruckner et al use the RAQMS-Aura chemical reanalysis to explore the relationship between tropospheric O3 and ENSO. Using observations from the TRMM satellite, they demonstrate that their reanalysis captures the variability in tropical convection. Then, using composite analysis, EOFs, and a multiple linear regression they show that ENSO is the dominant driver of O3 variability in the tropics. The MLR analysis highlights that ENSO-related dynamical changes are the dominant driver of this ENSO-related variability, although biomass burning also has a meaningful contribution. This paper is suitable for publication in ACP after the following minor issues are addressed.

Line 95: What AIRS CO products are you using? The total column, which is not advised, or the layers product? If it is the layers product, please describe which layers are included.

We are assimilating the AIRS Level 2 Support Retrieval Version 6 CO profile product (AIRS Science Team/Joao Teixeira, 2013), with accounting for vertical weighting functions as specified in Maddy and Barnet (2008).

The manuscript has been revised to state that we are using the AIRS CO profile product and to include reference to Maddy and Barnet (2008).

Section 3.1: While you perform a thorough evaluation of the RAQMS-Aura precipitation, a discussion on the evaluation of the accuracy of the O3 and CO using independent data is also warranted. While the anomaly plots in Section 3.2 do suggest that the reanalysis captures the ENSO related variability of the column amounts of these species, a more thorough analysis akin to the TRMM analysis would be beneficial. If this has already been done in a separate paper, a simple reference would suffice. Also, how accurate is the vertical O3 in the reanalysis? This could be evaluated with something like SHADOZ ozonesondes.

We agree that we should discuss the evaluation of the RAQMS-Aura O3 and CO analyses. A new section has been added to this manuscript covering evaluation with SHADOZ ozonesondes, OMI-MLS TOR, and MOPITT CO. [See lines 220-290 in the revised manuscript]

Figures 4 -6: Why not combine these into a single figure?

We thank the reviewer for this suggestion and have combined the timeseries presented in Figures 4-6 into a single Figure 4 with panels a, b, c corresponding to the figure numbers in the original submission.

Line 332: Show how you are defining ozone production. What reactions/species are included here?

We thank the reviewer for this suggestion and have added the following text describing RAQMS ozone production.

RAQMS has standard hydrogen oxides (HOx), chlorine oxides (ClOx), bromine oxides (BrOx), and NOx ozone photochemistry (Eckman et al., 1995) with Carbon Bond-Z (CB-Z) (Zaveri and Peters, 1999) treatment of nonmethane hydrocarbon chemistry. Chemical production and loss are calculated explicitly for the Ox family, which in RAQMS includes O(1D), O(3P), O3, NO2, HNO3, NO3, N2O5, HNO4, PAN (peroxynitrates), and MPAN.

Line 410: In your MLR analysis, you include both the PC1CO and PC1precip terms, which are somewhat related (r = -.435). How does this relationship affect the interpretability of the results of your MLR? How can you be sure that the impacts of one variable aren't being convolved with the impacts of the other?

Precipitation and CO variability are not independent, as biomass burning is suppressed by precipitation. To address how the co-variability in CO and precipitation affect the MLR, we have added a discussion of this co-variablity by removing each component of the regression independently and evaluate the change on the overall fit [see lines 514-523 in the revised manuscript]. We find that there is a slight improvement in the RMSE and R2 metrics by including CO PC1 as opposed to the fit when CO PC1 is not considered.

Section 3.3.2: Oman et al (2013) demonstrated an impact of the QBO on upper tropospheric O3. Can you separate the QBO and ENSO impacts on your results, since the Nino3.4 and QBO EOF1 indices are relatively strongly anti-correlated? How does omission of this term from your MLR affect the results? Some discussion of potential QBO impacts on your results is warranted.

We agree that there should be some discussion of the QBO impact on upper tropospheric O3 and have added a supplemental section that compares our RAQMS-Aura analysis to the Oman et al. 2013 analysis. RAQMS-Aura does include modulation of O3 by the QBO, as confirmed through evaluating tropical zonal mean zonal winds and zonal mean O3 timeseries (not shown). In the supplemental, we compare the RAQMS-Aura QBO signature to Oman et al (2013) and find that, during the 2006-2016 period of the RAQMS-Aura re-analysis the QBO contribution to upper tropospheric ozone is less than found by Oman et al 2013 based on MLS data during the period from 2004-2012. Omission of the QBO terms does not have a significant impact on the linear regression as the linear regression is evaluating influences on tropical tropospheric ozone ENSO variability, and the QBO has a small impact on upper tropospheric ozone.

**Response to Referee #2**

This study takes a closer look at ENSO related variability in tropical tropospheric ozone from the RAQMS-Aura chemical reanalysis. Using multiple linear regression analysis and compositing techniques to show the dominant drivers causing this ozone change depending on region. While the calculated response is similar to what has been shown in some past studies, the use of RAQMS-Aura appears novel including ozone production and loss terms, convective mass flux, and diabatic heating and should be of interest of ACP readership. I would recommend publication pending the authors consideration of a few mostly minor suggestions below.

I think it would be particularly helpful if the authors consider especially for figures 8-14, and 20 to show some indication of significance in the response plots to put the changes in context and help draw the readers eyes to the key areas.

We agree that adding shading to highlight significance in the composites would be useful and have updated the figures to shade where the composite is significant at the 95% confidence level from a t test.

Some of the ozone response to ENSO in the subtropics is related to stratosphere-troposphere exchange, are there any concerns about representing that properly with a model having 35 model levels. Does the use of assimilated data help with any model deficiencies related to resolving those processes?

The tropical upper tropopause does have lower vertical resolution than other regions because of the hybrid isentropic coordinates, which does introduce high biases in the upper tropopause ozone relative to ozonesondes. The assimilation of OMI and MLS improves the bias (Sekiya et al., in-prep). In our new supplemental section that looks at the response of upper troposphere tropical ozone to the QBO and ENSO which further evaluates the RAQMS-Aura ozone distribution in the tropical upper troposphere.

I don't believe you cite Oman et al. 2013, please have a look and note some of the discussion related to Quasi-biennial Oscillation (QBO) and its influence that can extend into the troposphere and consider checking and possibly adding terms to your regression analysis to account for its influence.

The revised manuscript now cites Oman et al. 2013. We have also included a supplemental section that reproduces the MLR analysis by Oman et al. 2013. We find that RAQMS-Aura captures the westerly-easterly phase shift in zonal mean equatorial zonal mean winds and the associated QBO impacts on stratospheric ozone very well. We also find that the QBO modulates upper tropospheric tropical ozone during the 2006-2016 RAQMS-Aura period in a manner that is similar to that found by Oman et al. 2013 during 2004-2012 using MLS data. However, the QBO influence on tropical tropospheric ozone is largest in the upper troposphere and smaller than the ENSO influence. Adding terms to the regression analysis for the QBO EOFs did not result in a significant change to the R, root mean square error, or the regression coefficients for CO PCs 1-3 and precipitation PC.

Any concerns about the shift in bias noted in precipitation fields between 2009-2011, compared to observations, impacting ozone response, are there corresponding shifts in any of the ozone response terms.

We acknowledge the larger amplitude in region could lead to higher amplitude ozone response, but we don't see any significant discontinuities in the equatorial ozone anomalies associated with this shift in precipitation biases [new section 3.2.2].

Lines 404-405 and comment in general: Shifts in precipitation also impact clouds, would this impact be reflected in your net ozone production related to changes in photolysis rates. Or stated more generally on the net ozone productions term can you differentiate that caused by biomass burning emission changes with clouds and photolysis changes.

The RAQMS-Aura reanalysis uses FastJ2 to calculate photolysis rates and only responds to changes in atmospheric transmittance due to large-scale resolved clouds. Since the shifts in precipitation within the tropics are largely associated with shifts in convective clouds (fig. 1) we are not able to address any changes in net ozone production related to changes in photolysis rates. The convective cloud mass flux results in a re-distribution of both ozone and ozone precursors, which can change the net ozone production. This has been clarified in the revised manuscript [see lines 410-413 in the revised manuscript]

Lines 51-53 while La Niña responds in a generally opposite manner it is not quite symmetrical

Thank you for this comment. This point has been clarified in the revised manuscript.

Line 130 al missing period al.

We thank the referee for catching this error and have added the period.

Line 339 need to subscript 3 in O3

We thank the referee for catching this typesetting error and have subscripted as suggested.

AIRS Science Team/Joao Teixeira (2013), AIRS/Aqua L2 Support Retrieval (AIRS+AMSU) V006, Greenbelt, MD, USA, Goddard Earth Sciences Data and Information Services Center (GES DISC), Accessed: [June 25, 2024], 10.5067/Aqua/AIRS/DATA207

Oman, L. D., Douglass, A. R., Ziemke, J. R., Rodriguez, J. M., Waugh, D. W., & Nielsen, J. E. (2013). The ozone response to ENSO in Aura satellite measurements and a chemistry-climate simulation. Journal of Geophysical Research-Atmospheres, 118(2), 965-976. https://doi.org/10.1029/2012jd018546

Seikiya, T., Emili, E., Miyazaki, K., Inness, A., Qu, Z., Pierce, R.B., Jones, D., Worden, H., Cheng, W.Y.Y., Huijnen, V., & Koren, G. (2024). Assessing the relative impacts of satellite ozone and its precursor observations to improve global tropospheric ozone analysis using multiple chemical reanalysis systems. Manuscript in preparation.